# Stabilizing PPO via Latent-Space Regularization and KDE-Driven Exploration

**Meiyu Du** [1 2]  **Yuqing Gao** [1 3]  **Wei Wang** [1 2]

## Abstract

Proximal Policy Optimization (PPO) is widely used in continuous-control tasks, yet its performance is often highly sensitive to training dynamics when neural networks approximate the policy and value functions. This paper introduces SPPO, a drop-in augmentation that preserves PPO's clipped objective and network topology while stabilizing actor-critic geometry via three mechanisms: (i) a Central Kernel Alignment (CKA)-based constraint on critic representations, (ii) a no-flip regularizer on actor updates, and (iii) Kernel Density Estimation (KDE)-driven advantage shaping. Theoretical analysis shows that these components tighten bounds on one-step bootstrapping error, improve expected directional alignment of action updates, and ensure nondecreasing occupancy mass over high-novelty regions. Experiments on standard continuous-control benchmarks demonstrate consistent gains over PPO and recent PPO stabilization methods. Ablation studies further quantify the contribution and complementary effects of each component. Additional training-dynamics analyses indicate that SPPO reduces instability and oscillations in both actor and critic updates, improving training stability and final performance.

## 1. Introduction

Owing to its simple clipped objective, ease of implementation, and robustness across tasks, Proximal Policy Optimization (PPO) (Schulman et al., 2017) remains a widely used baseline reinforcement learning (RL) algorithm for continuous-control tasks (Huang et al., 2022b) and has been applied in engineering applications, such as structural inspection (Wang et al., 2026) and structural optimization tasks (Du et al., 2025). In practice, however, combining PPO with function approximation and bootstrapped targets can lead to high variance, performance regressions, and sensitivity to implementation details (Engstrom et al., 2019; Huang et al., 2022a). In value-based RL, the interaction of function approximation, bootstrapping, and off-policy updates is known as the "deadly triad", which causes instability in value estimates (Zhang et al., 2021; Fujimoto et al., 2018). The training behavior of PPO is also highly influenced by how the actor and critic networks fit the data rather than by the algorithmic objective alone. Beyond algorithmic refinements, it remains necessary to constrain training from the perspective of neural representations and latent-space geometry in order to improve agent performance in practical tasks.

To characterize the evolution of actor and critic representations during PPO training, Central Kernel Alignment (CKA) (Cortes et al., 2012) is used here to compare layer-

[1]State Key Laboratory of Disaster Reduction in Civil Engineering, Tongji University, Shanghai, 200092, China [2]Department of Structural Engineering, Tongji University, Shanghai, 200092, China [3]Department of Disaster Mitigation for Structures, Tongji University, Shanghai, 200092, China. Correspondence to: Yuqing Gao <yuqing27@tongji.edu.cn>.

Proceedings of the *43rd International Conference on Machine Learning*, Seoul, South Korea. PMLR 306, 2026. Copyright 2026 by the author(s).

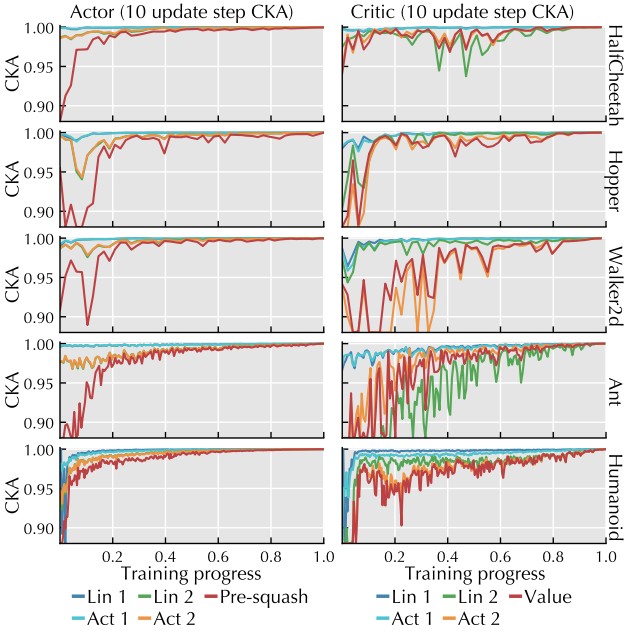

*Figure 1.* Evolution of CKA for intermediate actor and critic representations over the course of training across five environments.

wise subspaces across training snapshots (Kornblith et al., 2019). A fixed state pool is constructed by aggregating states encountered during training, and network parameters are snapshotted every 10 updates. For each snapshot pair and each layer, pre- and post-activation representations on this state pool are compared using linear CKA; formal definitions are provided in Appendix A. Fig. 1 reports the CKA trajectories during training of a single run in five MuJoCo environments (Todorov et al., 2012) using PPO with two hidden layers in both the actor and the critic. Fig. 9 provides results over six random seeds, the mean trajectory with the min-max envelope was plotted, and the aggregated results show the same qualitative pattern as in Fig. 1. Experimental details are given in Appendix B. The curves are named by the location of representations: "Lin 1" denotes the output of the first linear layer, and "Act 1" denotes the output after the first activation.

The CKA values are generally low in early training, indicating rapid reshaping of intermediate representations. As training progresses, most layers exhibit increasingly stable subspace structure, and the actor enters a stable CKA regime earlier. Notably, the last critic hidden layer before the output (Act 2 in the second column of Fig. 1) converges more slowly and often shows substantial subspace changes into mid and late training; a similar pattern is observed for the value head (Value in Fig. 1). For the actor, the last linear layer (Pre-Squash) converges more slowly. Appendix A reports results with three hidden layers, showing analogous trends with more pronounced effects. These observations suggest repeated geometric adjustment in the representation feeding the linear value head and continued non-smooth rearrangements in the directions of policy outputs.

In this paper, instability refers specifically to oscillation of the network update during training rather than performance fluctuation in a broad sense. Given a fixed set of probe states, instability occurs when two consecutive updates introduce unnecessary or non-smooth changes in the critic's value predictions, the critic's latent representations, or the actor's pre-squash mean actions. Representational drift describes how the critic's latent features change across updates on the same probe states. Flipping refers to a large directional reversal of the actor's current pre-squash mean-action change relative to the improvement direction on the same state. Such oscillation may increase actor-critic mismatch and weaken the consistency between value estimation and policy improvement.

Motivated by these findings, this paper introduces a latent-space stabilization scheme for PPO in continuous control, termed *Stabilized PPO* (SPPO). SPPO operates on actor and critic latent spaces via (i) a CKA-based constraint that stabilizes critic representations (Fig. 2(1)), (ii) a no-flip constraint that discourages sign flips of the actor's mean

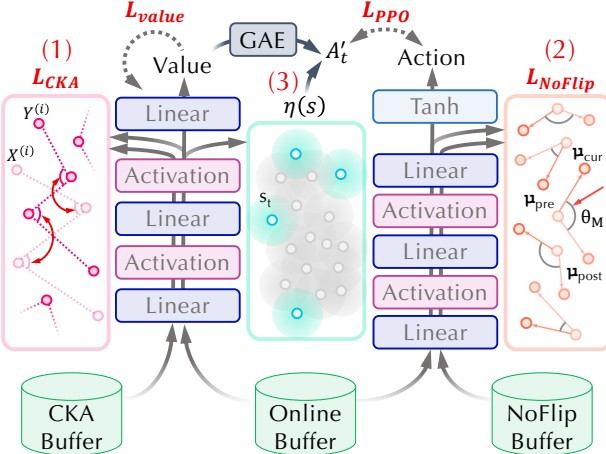

*Figure 2.* SPPO: (a) CKA-based constraint on critic representations; (b) no-flip constraint on actor pre-squash actions; (c) KDE-based advantage shaping in the critic latent space.

pre-squash action (Fig. 2(2)), and (iii) a kernel density estimation (KDE)-based adjustment of action advantages that encourages exploration toward low-density regions in the critic latent space (Fig. 2(3)). The combined effect of these three improvement measures stabilizes actor and critic network training while enhancing exploration, thereby further boosting performance. Sec. 2 reviews related work; Sec. 3 presents the three components, their integration into PPO, and theoretical properties; Sec. 4 reports experiments and analyses of stability improvements of SPPO in training; and Sec. 5 concludes with future directions.

## 2. Related works

During RL training, estimation bias and instability in function learning are regarded as major obstacles to stability and generalization (Zhang et al., 2021; Van Hasselt et al., 2018; Nauman et al., 2024). Early work improves value function stability through experience replay and target networks (Mnih et al., 2015; Kumar et al., 2020; Chen et al., 2023), and subsequent methods mitigate instabilities by modifying update rules (Van Hasselt et al., 2016; Hessel et al., 2018) or correcting bootstrapping bias and value overestimation (Fujimoto et al., 2018; Piché et al., 2021). Ensembling (Chen et al., 2021; Kuznetsov et al., 2020) and auxiliary regularization (Kumar et al., 2021; Pavse et al., 2025) have also been investigated. Other studies analyze error propagation (Tang & Berseth, 2024) and interference (Liu et al., 2023), as well as representation collapse (Igl et al., 2020; Lyle et al., 2022a;b; Sokar et al., 2023).

Recent work introduces explicit regularization on internal representations to counter degradation induced by bootstrapping and non-stationarity (Gulcehre et al., 2022; He et al., 2023). For PPO, Moalla et al. (Moalla et al., 2024)conduct

a systematic analysis of the causes of performance collapse and propose an $\ell_2$ regularizer on pre-activation features of the policy network. At the policy level, Trust Region Policy Optimization (TRPO) and PPO constrain the policy update to avoid drastic behavioral changes (Schulman et al., 2015a; 2017), and recent work has revealed policy churn, where frequent policy changes undermine behavioral consistency and accumulate critic estimation errors (Schaul et al., 2022). Churn has also been linked to loss of plasticity via gradual NTK rank collapse, and explicitly reducing churn can help preserve adaptability in continual RL (Tang et al., 2025) . Multi-stage policy-gradient methods (Cobbe et al., 2021) and regularizers that align the current policy with historical policies (Tang & Berseth, 2024) have been proposed to reduce such oscillations. Under dynamic shift, policy regularization on globally accessible states has been introduced to improving cross-dynamics transfer (Xue et al., 2025) .

Exploration in RL has been promoted through optimistic initialization, which assigns high initial values to unvisited states (Machado et al., 2015), and through intrinsic rewards, including random network distillation (Burda et al., 2018), density estimation(Bellemare et al., 2016; Seo et al., 2021; Domingues et al., 2021; Hazan et al., 2019), generative modeling (Liu & Liu, 2025) and self-supervised predictive models (Guo et al., 2022; Jarrett et al., 2022). These studies indicate that the stability and final performance of deep RL algorithms depend not only on accurate value estimation and exploration design, but also on the geometry and dynamics of internal representations and on how policy updates are constrained.

This study places regularization of the critic in the latent space between nonlinear activations and the value head, directly shaping the representation space in which value semantics are expressed while allowing the value head to approximate the value distribution with sufficient freedom. On the actor side, a no-flip constraint is introduced that acts in the pre-squash space and is integrated as an auxiliary regularizer into mainstream policy-gradient algorithms without modifying the primary optimization objective. The KDE driven exploration operates in the latent spaces already present inside the critic, enforcing representational consistency and shaping exploration bonuses based on latent-space sparsity so as to alleviate training issues caused by representational inconsistency. From a functional perspective, the first two improvements are used to directly stabilize the actor and critic networks, while KDE-driven exploration is employed to counteract potential under-exploration issues that may arise under regularization.

## 3. SPPO

### 3.1. Preliminaries

SPPO build on PPO applied to continuous-control tasks, modeled as a finite-horizon Markov decision process (MDP) $M = (\mathcal{S}, \mathcal{A}, P, r, \gamma, \rho_0)$, where $\mathcal{S}$ is the state space, $\mathcal{A}$ is the action space, $P$ is the transition kernel, $r$ is the reward function, $\gamma \in (0, 1)$ is the discount factor, and $\rho_0$ is the initial state distribution (Sutton et al., 1998).

For any state $s \in \mathcal{S}$, the policy $\pi_\phi(a \mid s)$ is represented by a neural network parametrized by $\phi$ that outputs a mean vector $\mu_\phi(s)$, from which a Gaussian variable is sampled and then passed through a $\tanh$ squashing function to obtain the action:

$$u \sim \mathcal{N}\big(\mu_\phi(s), \Sigma\big), \qquad a = \tanh(u) \in (-1, 1)^d,$$

where $\Sigma = \mathrm{diag}(\sigma^2)$ is a diagonal covariance matrix and $\sigma \in \mathbb{R}^d$ is a vector of state-independent, learnable standard deviations. The value network can be written as the composition of an encoder and a linear head,

$$h_\theta : \mathcal{S} \to \mathbb{R}^H, \qquad V_{\theta,\omega}(s) = \omega^\top h_\theta(s),$$

where $h_\theta$ denotes the encoder representation with trainable parameters $\theta$, $\omega \in \mathbb{R}^H$ is the parameter vector of the final linear layer, and $V_{\theta,\omega}(s)$ is the value estimate.

The three components proposed in SPPO are introduced in this section, together with verifiable properties. Full derivations are deferred to Appendix D. It should be noted that the derivations in this section and Appendix D are not intended as a strict global description of full deep nonlinear training dynamics, but as a local mechanistic analysis of how the proposed regularizers can improve conditioning and tighten upper bounds on representation drift and bootstrapping error under local approximation. The more important question is whether these mechanisms have observable empirical counterparts, which are discussed in detail in Sec. 4.3.

### 3.2. CKA-based Critic Latent Regularization

In the critic network, the encoder $h_\theta(\cdot)$ maps states from $\mathcal{S}$ into a latent space on which the linear head $\omega^\top h_\theta(s)$ defines a scalar field of value estimates. Update can be viewed as simultaneously adjusting the shape of the latent representation and the scalar readout. As the policy changes, though the value of off-policy samples may drift, their relative relationships and local geometry within the same short sequence should remain consistent within a few updates. This is encouraged in SPPO by imposing a linear CKA alignment on the critic latent representations of historical short sequences.

During training, a frozen copy of the critic encoder from the previous update, denoted by $\bar{h}_\theta(\cdot)$, is maintained together with a buffer $\mathcal{B}_{CKA}$ that stores past state trajectories.

At each critic update, $m$ short sequences of length $L$ consisting of consecutive time steps are sampled from $\mathcal{B}_{CKA}$. For the $i$-th short sequence, the old and new latent representation matrices $X^{(i)} \in \mathbb{R}^{L \times H}$ and $Y^{(i)} \in \mathbb{R}^{L \times H}$ are constructed by stacking encoder outputs along the temporal dimension, where $H$ is the dimensionality of the latent space. For a given pair $(X^{(i)}, Y^{(i)})$, the linear CKA value $\mathrm{CKA}(X^{(i)}, Y^{(i)}) \in [0,1]$ is defined as in Appendix A (cf. Eq. (1)). This yields the structure-preserving regularizer:

$$\mathcal{L}_{\mathrm{CKA}}(\theta; \bar{\theta}; \mathcal{B}) = \frac{1}{m} \sum_{i=1}^{m} \big(1 - \mathrm{CKA}(X^{(i)}, Y^{(i)})\big),$$

where $\bar{\theta}$ is the parameters of the frozen encoder $\bar{h}_\theta$. The old latent matrices $X^{(i)}$ are treated as constants. The CKA regularizer tightens bounds on distribution sensitivity and bootstrapping error. A single value-function update can be written as a ridge-regression problem on a sample set $\mathcal{B} = \{(s_i, y_i)\}_{i=1}^{n}$, where $y_i$ denotes TD/GAE targets. Define

$$A_0 = X^\top X + \lambda I, \quad A_{\mathrm{CKA}} = X^\top X + \lambda I + M_{\mathrm{CKA}},$$

where $X \in \mathbb{R}^{n \times d}$ is the design matrix, and $r \in \mathbb{R}^n$ is the residual vector. The CKA regularizer constructed from old short sequences is equivalent to adding a positive semidefinite matrix $M_{\mathrm{CKA}} \succeq 0$ to the normal matrix, and $A_{\mathrm{CKA}} \succeq A_0$ holds in the Löwner order.(Appendix D.1.2)

**Proposition 3.1 (Tightened bound on pointwise latent drift).** For any state $s'$, the change in its latent representation $\delta h_\theta(s')$ after a small update satisfies

$$\begin{aligned}
\big\|\delta h_\theta(s')\big\| &\leq \big\|J_\theta(s')\big\| \big\|A_{\mathrm{CKA}}^{-1} X^\top\big\| \|r\| \\
&\leq \big\|J_\theta(s')\big\| \big\|A_0^{-1} X^\top\big\| \|r\|,
\end{aligned}$$

where $J_\theta(s') = \partial h_\theta(s')/\partial \theta$ is the Jacobian of the encoder at $s'$. The CKA term lifts the spectrum of the normal matrix and tightens the upper bound on pointwise representation drift.(Appendix D.1.3).

**Proposition 3.2 (Tightened bound on mean-squared drift over unvisited states).** Let the set of unvisited states be $\mathcal{U} = \{s'_j\}_{j=1}^m$, and stack their Jacobians into a block Jacobian $J_u$ and the changes in their latent representations into a matrix $\Delta H_u$. Under the assumption that the residual noise has bounded covariance, the following bound holds:

$$\begin{aligned}
\mathbb{E}\big[\|\Delta H_u\|_F^2\big] \\
&\leq \lambda_{\max}\big(\mathrm{Cov}(r)\big) \, \mathrm{Tr}\big(J_u A_{\mathrm{CKA}}^{-1} X^\top X A_{\mathrm{CKA}}^{-1} J_u^\top\big) \\
&\leq \lambda_{\max}\big(\mathrm{Cov}(r)\big) \, \mathrm{Tr}\big(J_u A_0^{-1} X^\top X A_0^{-1} J_u^\top\big),
\end{aligned}$$

where $\lambda_{\max}(\cdot)$ is the largest eigenvalue. In the mean-squared sense, representation drift over the unvisited set is no larger, and typically smaller, after adding the CKA regularizer (Appendix D.1.3).

**Corollary 3.3 (Contraction of a second-order upper bound on one-step bootstrapping error).** Let $V^\pi$ denote the ideal value function, and define the estimation error $e(s) = \hat{V}(s) - V^\pi(s)$ and the one-step bootstrapping error $\varepsilon_{\mathrm{boot}}(s, s') = \gamma e(s')$. After an update,

$$\begin{aligned}
\mathbb{E}\big[\varepsilon_{\mathrm{boot,new}}^2\big] &= \gamma^2 \mathbb{E}\big[e_{\mathrm{new}}(s')^2\big] \\
&\leq 2\gamma^2 \mathbb{E}\big[e_{\mathrm{old}}(s')^2\big] + C \big\|A_{\mathrm{CKA}}^{-1} X^\top\big\|^2 \|r\|^2,
\end{aligned}$$

for some constant $C > 0$. Compared with the case without CKA, the inequality $\|A_{\mathrm{CKA}}^{-1} X^\top\| \leq \|A_0^{-1} X^\top\|$ ensures that this second-order upper bound on the one-step bootstrapping error is tighter.(Appendix D.1.4)

In implementation, the CKA term is multiplied by a weighting hyperparameter $\lambda_{\mathrm{CKA}} > 0$ and added to the value loss. It exploits the relative structure of historical short sequences to resist distortions in the latent space induced by the coexistence of distributional shift and bootstrapping.

### 3.3. No-Flip Penalty

The CKA curve in Fig. 1 indicate that the actor's pre-squash space still undergoes substantial structural rearrangements in the middle and late training stages. The PPO objective can be viewed as updating the mean action through a combination of a probability ratio and an advantage term with clipping. The clipping term prevents excessively large updates, but does not explicitly avoid pushing historical samples in directions opposite to their previous improvement directions. Thus, a directional-consistency (no-flip) regularizer is introduced. For past samples, the historical improvement direction is recorded, and a penalty is applied when the direction of the current mean-action change in the pre-squash space forms an excessively large angle with this historical direction.

A history buffer $\mathcal{B}_{NoFlip}$ is thus maintained, containing tuples $(s, \mu_{\mathrm{pre}}(s), \mu_{\mathrm{post}}(s))$, where $\mu_{\mathrm{pre}}(s)$ is the mean (in pre-squash space) before a certain actor update and $\mu_{\mathrm{post}}(s)$ is the mean after that update. During the current actor update with parameters $\phi$, a set of states $\{s_i\}_{i=1}^M$ is sampled from $\mathcal{B}_{NoFlip}$, and the current mean $\mu_\phi(s_i)$ and covariance $\Sigma_{\mathrm{cur}}$ are evaluated. For each sample $s$, define the old and new update directions

$$d_{\mathrm{old}}(s) = \mu_{\mathrm{post}}(s) - \mu_{\mathrm{pre}}(s), \quad d_{\mathrm{new}}(s) = \mu_\phi(s) - \mu_{\mathrm{pre}}(s).$$

Using the Mahalanobis geometry induced by $\Sigma_{\mathrm{cur}}^{-1}$, the cosine between $d_{\mathrm{new}}(s)$ and $d_{\mathrm{old}}(s)$ is

$$\cos\theta_M(s) = \frac{d_{\mathrm{new}}(s)^\top \Sigma_{\mathrm{cur}}^{-1} d_{\mathrm{old}}(s)}{\sqrt{d_{\mathrm{new}}(s)^\top \Sigma_{\mathrm{cur}}^{-1} d_{\mathrm{new}}(s)} \sqrt{d_{\mathrm{old}}(s)^\top \Sigma_{\mathrm{cur}}^{-1} d_{\mathrm{old}}(s)} + \varepsilon},$$

where $\varepsilon$ is a small positive constant for numerical stability. Let the angle threshold be $\alpha_{\mathrm{thr}} \in [0, \pi/2]$ and define $c_{\mathrm{thr}} =$

$\cos(\alpha_{\text{thr}})$. The no-flip penalty is defined by a hinge loss

$$\mathcal{L}_{\text{noflip}}(\phi; \mathcal{B}_{NoFlip}) = \frac{1}{M} \sum_{i=1}^{M} \big[ c_{\text{thr}} - \cos\theta_M(s_i) \big]_+,$$

where $[x]_+ = \max(0, x)$. When $\cos\theta_M(s) > c_{\text{thr}}$, the penalty is zero; otherwise, the penalty is activated, suppressing updates that push the mean action of a historical sample into a direction forming an excessively large angle with its historical improvement direction. In this study $\alpha_{\text{thr}}$ takes the value of $\pi/2$ and thus formed the "No-Flip" penalty.

The No-Flip regularizer improves expected alignment and controls the strong-violation probability. Define

$$M_1(\phi) = \mathbb{E}_{s \sim \mathcal{D}_{\text{hist}}} \big[ \cos\theta_M(s) \big],$$
$$M_2(\phi, \delta) = \mathbb{P}\big( \cos\theta_M(s) \leq c_{\text{thr}} - \delta \big),$$

which measure, respectively, the expected directional alignment and the tail probability of strong violations where the angle is larger than $\alpha_{\text{thr}}$. (Appendix D.2)

**Lemma 3.4 (Expected-cosine lower bound and strong-violation tail bound).** For any $\phi$ and any $\delta > 0$,

$$M_1(\phi) \geq c_{\text{thr}} - \mathcal{L}_{\text{noflip}}(\phi),$$
$$M_2(\phi, \delta) = \mathbb{P}\big( \cos\theta_M(s) \leq c_{\text{thr}} - \delta \big) \leq \frac{\mathcal{L}_{\text{noflip}}(\phi)}{\delta}.$$

A smaller $\mathcal{L}_{\text{noflip}}(\phi)$ implies a larger expected alignment and yields a tighter upper bound on the probability of strong-violation events.(Appendix D.2.1)

Let the learning rate be $\alpha > 0$ and define the original PPO descent direction $v_{\text{PPO}}(\phi) = -\nabla_\phi \mathcal{L}_{\text{PPO}}(\phi)$. With the no-flip regularizer added, the combined descent direction becomes

$$v_{\text{total}}(\phi) = -\nabla_\phi \mathcal{L}_{\text{PPO}}(\phi) - \lambda_{\text{noflip}} \nabla_\phi \mathcal{L}_{\text{noflip}}(\phi),$$

with weight $\lambda_{\text{noflip}} > 0$.

**Proposition 3.5 (First-order improvement in alignment and tail control).** Under a first-order expansion with small step size $\alpha$, the no-flip loss satisfies

$$\mathcal{L}_{\text{noflip}}(\phi + \alpha v_{\text{total}}) = \mathcal{L}_{\text{noflip}}(\phi + \alpha v_{\text{PPO}}) - \alpha\lambda_{\text{noflip}} \big\| \nabla_\phi \mathcal{L}_{\text{noflip}}(\phi) \big\|^2 + o(\alpha).$$

Combining this relation with Lemma 3.4 yields, to first order, tighter bounds on the expected $M_1(\phi)$ and $M_2(\phi, \delta)$ under the combined update than under the original PPO update. (Appendix D.2.2).

Notably, gradients from the no-flip term are propagated only through the current actor parameters $\phi$; the historical means $\mu_{\text{pre}}$ and $\mu_{\text{post}}$ are used solely to construct directions.

### 3.4. KDE-Driven Novelty Shaping

Exploration in PPO is primarily driven by policy stochasticity. Beyond visiting high-value states, reaching novel states broadens the coverage of online data over the reachable state space, which can mitigate overfitting and reduce extrapolation error. Measuring density or distance directly in the raw state space often lacks semantic meaning, and previous approaches have instead used encoders to obtain representations before quantifying distances. However, the critic's latent representations are already trained under value-prediction objectives and thus carry value-related semantics. Estimating density or novelty in this latent space therefore yields a notion of novelty that is more closely aligned with the learning objective.

Let the set of online states be $\mathcal{S}_{\text{on}} = \{s_t\}_{t=1}^{n}$. An KDE can be constructed in the critic latent space,

$$\hat{p}\big(h_\theta(s)\big) = \frac{1}{n} \sum_{j=1}^{n} K\big(h_\theta(s), h_\theta(s_j)\big),$$

where $n = |\mathcal{S}_{\text{on}}|$, $\eta(s) = -\log \hat{p}\big(h_\theta(s)\big)$, and $K$ is a Gaussian kernel with bandwidth $b > 0$ on $\mathbb{R}^H$. The quantity $\eta(s)$ is used as a novelty score. To avoid introducing a global bias and to preserve the original advantage signal, a quantile-based baseline is applied to $\{\eta(s_t)\}_{t=1}^{n}$:

$$\tau = q_p\big(\{\eta(s_t)\}_{t=1}^{n}\big), \quad \tilde{\eta}(s) = \max\big(0, \eta(s) - \tau\big),$$

where $q_p$ denotes the $p$-quantile. $\tilde{\eta}(s)$ is used to shape the advantage of the action that leads into a novel state. If the transition $s_t \to s_{t+1}$ is observed, the shaped advantage is defined as

$$\hat{A}'_t = \hat{A}_t + \lambda_{\text{nov}} \tilde{\eta}(s_{t+1}),$$

where $\hat{A}_t$ is the GAE($\lambda$) advantage estimate (Schulman et al., 2015b) and $\lambda_{\text{nov}} > 0$ is a weighting hyperparameter. In the clipped PPO surrogate on the actor side, only $\hat{A}_t$ is replaced by $\hat{A}'_t$, and this surrogate is combined with the no-flip term to form the complete actor loss $\mathcal{L}_{\text{actor}}(\phi)$.

The KDE-driven novelty shaping improves occupancy quality in the value latent space. Consider a state–action occupancy measure $\rho$ induced by the behavior policy and environment dynamics. For any measurable function $f$, write $\langle \rho, f \rangle = \int f(x) \, d\rho(x)$. Let $g(x) \geq 0$ denote a novelty field defined in the latent space. Adding a term proportional to $\langle \rho, g \rangle$ to the policy objective is equivalent, in occupancy space, to applying an exponential tilting to $\rho$:

$$d\rho^+(x) \propto \exp\big(\beta(g(x) - b)\big) \, d\rho(x),$$

where $\beta > 0$ controls the strength of the shaping and $b$ is a baseline.

Define the high-novelty (low-density) region $A = \{x : g(x) \geq \tau_0\}$, and its occupancy mass under $\rho$ and $\rho^+$ as $m =$

$\langle \rho, \mathbb{1}_A \rangle$ and $m^+ = \langle \rho^+, \mathbb{1}_A \rangle$, where $\mathbb{1}_A$ is the indicator function of $A$.(Appendix D.3.2)

**Theorem 3.6 (Non-decreasing occupancy mass in high-novelty regions).** If the baseline satisfies $b \leq \tau_0$, then for any $\beta > 0$,

$$m^+ \geq m.$$

When both the high-novelty region $A$ and its complement have nonzero measure under $\rho$, and $\beta > 0$ is nontrivial, the inequality is strict, and exponential tilting increases visitation probability of low-density (high-novelty) regions in generic cases.(Appendix D.3.3). The truncated novelty signal $\tilde{\eta}(s_{t+1})$ can be viewed as a truncated version of $g(x)$ constructed in the critic latent space.

The CKA and no-flip regularizer act on the latent space geometry of the critic and actor, respectively, reducing instabilities, whereas the KDE-driven novelty shaping guides the occupancy distribution for wider coverage. The three components are mutually independent and compatible with the PPO surrogate, providing complementary control over representation stability and occupancy quality.

## 4. Experiments

This section evaluates SPPO on continuous-control tasks from MuJoCo-v5 (Todorov et al., 2012) and the Deep-Mind Control Suite (DMC) (Tassa et al., 2018), and assesses its applicability on five pixel-based, discrete-action Atari 2600 games from the Arcade Learning Environment (ALE) (Bellemare et al., 2013). On MuJoCo-v5, three closely related algorithms, namely CHAIN (Tang & Berseth, 2024), PEER (He et al., 2023), and PFO (Moalla et al., 2024), are used as baselines alongside PPO. PPO, SPPO, and the three baseline algorithms share identical network architectures, optimizers, and general hyperparameter settings. By default, both the actor and the critic are implemented with two hidden layers of 128 units. The present comparison is intended as a controlled comparison rather than an exhaustive tuning comparison.

The experiments focus on three main questions: (i) how SPPO performs across network architectures with varying depth and width; (ii) the individual and combined effects of the three components; and (iii) how these components alter training dynamics. All reported results are based on six random seeds. Details of implementations are provided in Appendix B. Extra results and additional sensitivity analyses of key internal hyperparameters in SPPO are in Appendix C.

### 4.1. Performance and Scaling with Network Depth and Width

Fig. 3 shows training curves of PPO and SPPO in five MuJoCo-v5 environments under different network capaci-

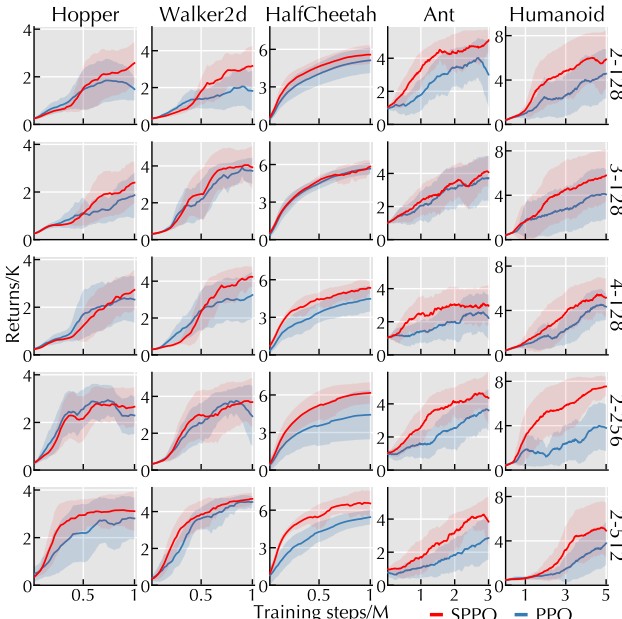

*Figure 3.* Comparison of PPO and SPPO under varying network capacities on MuJoCo-v5.

ties. The number and width of hidden layers are marked on the right of each row. The final evaluation returns of the five algorithms with default network configuration are provided in Tab. 1. These statistics are computed over six random seeds, where each seed is summarized by the mean return over the last ten evaluation points. Full results can be found in Appendix C.

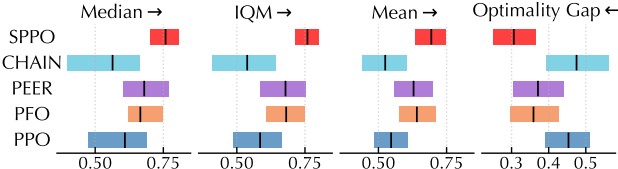

*Figure 4.* RLiable metrics for the five algorithms on MuJoCo-v5.

Across most environments and network capacities, SPPO outperforms PPO and the other baselines. SPPO also improves sample efficiency, as it achieves higher returns earlier in many settings (Fig. 3). The performances of the five algorithms across all environments, network capacities, and random seeds are further aggregated and evaluated using RLiable (Agarwal et al., 2021). Median, inter-quartile mean (IQM), mean, and optimality gap (as defined in RLiable) are reported as aggregate metrics with 95% confidence intervals, and the results are summarized in Fig. 4. SPPO exhibits a stable performance gain over PPO and remains competitive with PEER and PFO. CHAIN tends to show pronounced gains only at larger network capacities (see Fig. 13). Detailed comparisons for each network configuration are reported in Appendix C.2.

*Table 1.* Final evaluation returns for MuJoCo tasks (2×128).

| ALGO | HOPPER | WALKER2D | HALFCHEETAH | ANT | HUMANOID |
|---|---|---|---|---|---|
| SPPO | **2436.48 ± 920.34** | 3174.42 ± 1039.89 | **5571.29 ± 775.65** | **4858.68 ± 1047.94** | **5842.28 ± 2684.43** |
| CHAIN | 862.27 ± 250.08 | 1182.63 ± 418.31 | 2898.99 ± 833.20 | 4153.74 ± 828.51 | 5715.42 ± 2031.42 |
| PEER | 2007.34 ± 905.35 | 2853.52 ± 1082.23 | 3894.80 ± 1419.79 | 4226.03 ± 1387.23 | 4740.86 ± 2218.58 |
| PFO | 1529.09 ± 384.68 | **3256.65 ± 1020.90** | 4614.90 ± 1159.97 | 4825.22 ± 783.84 | 5526.12 ± 2412.89 |
| PPO | 1633.21 ± 731.40 | 1923.51 ± 1092.51 | 5094.43 ± 1013.03 | 3117.58 ± 2043.12 | 4564.38 ± 2311.94 |

For DMC (Tassa et al., 2018), five environments are selected to compare the training curves and RLiable metrics as shown in Fig. 5. The final evaluation returns are provided in Tab. 2. SPPO's higher returns suggest that its benefits are not limited to MuJoCo and generalize across continuous-control benchmarks.

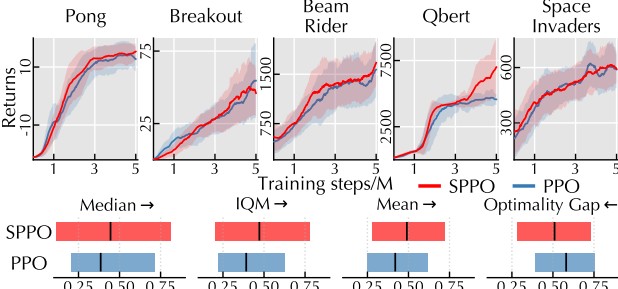

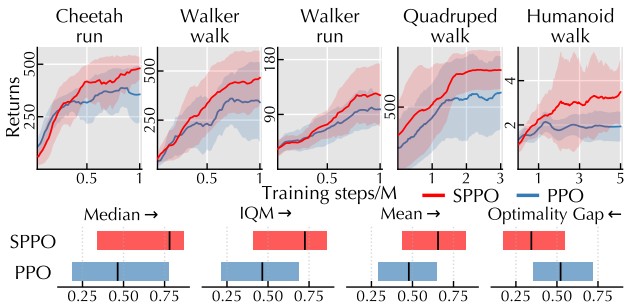

*Figure 5.* Training curves and RLiable metrics on DMC.

*Table 2.* Final evaluation returns for DMC and Atari environments.

| ENV | PPO | SPPO |
|---|---|---|
| CHEETAH-RUN | 344.4 ± 156.6 | **475.8 ± 74.3** |
| WALKER-WALK | 351.6 ± 199.6 | **447.2 ± 156.8** |
| WALKER-RUN | 99.9 ± 30.2 | **124.9 ± 50.8** |
| QUADRUPED-WALK | 651.8 ± 312.2 | **805.1 ± 155.9** |
| HUMANOID-WALK | 1.9 ± 0.7 | **3.3 ± 1.4** |
| PONG | 13.1 ± 4.6 | **15.2 ± 3.5** |
| BREAKOUT | **58.9 ± 60.9** | 46.8 ± 19.4 |
| BEAMRIDER | 1551 ± 563 | **1671 ± 337** |
| QBERT | 4630 ± 764 | **7292 ± 2340** |
| SPACEINVADERS | **602.8 ± 151.1** | 591.0 ± 94.2 |

To examine applicability beyond continuous control, five pixel-based, discrete-action environments from the Atari suites (Bellemare et al., 2013) are considered. The No-Flip regularizer is applied to the logits. Training curves and RLiable metrics are shown in Fig. 6, and the final evaluation returns are provided in Tab. 2.

Since these tasks involve discrete actions and image-based observations, the CKA and no-flip regularizers are applied as heuristic regularizers rather than as task-specific constraints. SPPO improves over PPO primarily in the later stages of training on Qbert, while the two methods perform comparably on the other environments. Although SPPO

*Figure 6.* Training curves and RLiable metrics on Atari.

is not designed specifically for pixel-based discrete-action tasks, it does not appear to harm performance in these settings.

The runtimes of SPPO and PPO on the five MuJoCo-v5 environments are reported in Table 3. The additional runtime mainly arises from KDE estimation. Algorithmically, KDE is applied in the critic latent space rather than the raw state space, and therefore is not directly determined by the raw state dimension. However, its efficiency is still affected by the latent dimension and the number of online samples. Future work may explore KDE estimation in a lower-dimensional value-semantic space to further reduce the additional computational overhead.

*Table 3.* Runtimes of SPPO and PPO on MuJoCo-v5.

| ENV | SPPO | PPO |
|---|---|---|
| HOPPER | 35 MIN | 21 MIN |
| WALKER2D | 36 MIN | 22 MIN |
| HALFCHEETAH | 33 MIN | 24 MIN |
| ANT | 95 MIN | 65 MIN |
| HUMANOID | 172 MIN | 115 MIN |

### 4.2. Component Contributions and Synergistic Effects

The compared variants include SPPO (all three components enabled), *w/o X* (disable component X only), and *Only X* (enable component X only). All variants share the same training setup as in Sec. 4.1. Fig. 7 shows the RLiable metrics evolve over training, and full training-curve comparisons are provided in Appendix C.3.

The results indicate that each component, when used solely,

yields noticeable improvements over PPO. Regarding their combined effects, the CKA-based regularization contributes primarily to stabilization and performance gains in the middle and late training stages, as evidenced by the separation of the red and orange curves in the first row of Fig. 7 later in training. The No-Flip term affects performance throughout the full training process, as the red and orange curves in the second row separate at the beginning and remain consistent as training progresses. By contrast, KDE alone yields a milder gain, mainly in the early and middle stages, suggesting it is better viewed as an exploration-enhancing component rather than the main source of overall stabilization. More importantly, when KDE is combined with only one of the other two components, its gain remains limited. An explanation consistent with the observed results is that KDE changes exploration preferences through advantage shaping, but when only one side of the actor-critic pair is explicitly stabilized, this exploration benefit may not be effectively absorbed by the other side, and can therefore be partially offset. In contrast, when both CKA and No-Flip are present, the critic and actor are both explicitly stabilized, so the exploration signal introduced by KDE can be absorbed more effectively. This pattern also suggests that the influence of KDE-based shaping may benefit from being adapted to the training dynamics, which is left for future work.

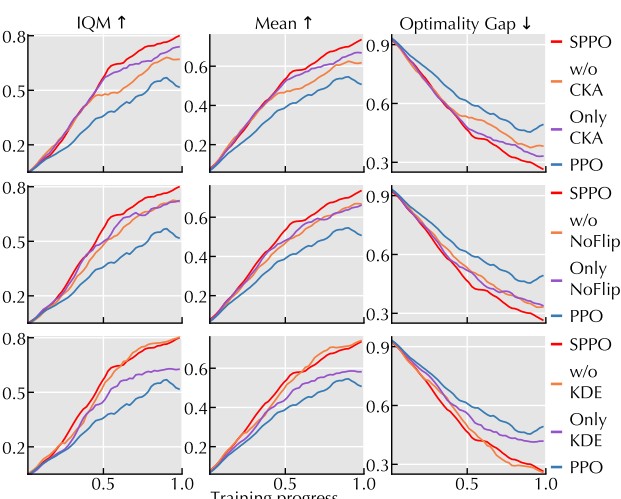

*Figure 7.* Ablation study of the three SPPO components.

The combined gains are not purely additive because the components exhibit nontrivial interactions, consistent with coupling through representation stability, update consistency, and exploration behavior, yielding more coherent training dynamics.

## 4.3. Mechanistic Explanation via Representation, Value, and Policy Dynamics

To compare SPPO and PPO in terms of network training dynamics, parameter snapshots of both the actor and the critic are stored at each update, and all states encountered during training are collected to approximate the reachable-state distribution. From this state pool, a fixed set of $K$ representative states $\{s_i\}_{i=1}^{K}$ ($K = 2048$ in this study) is sampled uniformly and used as a static probe set. At each update, the probe states are fed through the actor and critic to track intermediate representations, value estimates, and action outputs.

Six metrics are constructed to quantify changes between consecutive updates, as summarized in Tab. 4. The formulation and explanations of the metrics are provided in Appendix C.4. These metrics capture complementary aspects of training dynamics and churn (Schaul et al., 2022), and the curves in Fig. 8 are aggregated over six random seeds. Due to space constraints, results are shown for three representative environments, and additional environments are reported in Appendix C.5.

*Table 4.* Metrics for network training dynamics and churn between consecutive updates.

| # | METRIC NAME | DEFINITION |
|---|---|---|
| 1 | VALUE CHANGE | $\frac{1}{K}\sum_{i=1}^{K}\big(V_t(s_i) - V_{t-1}(s_i)\big)$ |
| 2 | VALUE CHURN | $\frac{1}{K}\sum_{i=1}^{K}\big|V_t(s_i) - V_{t-1}(s_i)\big|$ |
| 3 | CKA | $\mathrm{CKA}(H_t, H_{t-1})$ |
| 4 | CKA CHURN | $1 - \mathrm{CKA}(H_t, H_{t-1})$ |
| 5 | POLICY L2 | $\frac{1}{K}\sum_{i=1}^{K}\big\|a_t(s_i) - a_{t-1}(s_i)\big\|_2$ |
| 6 | POLICY COS | $1 - \frac{1}{K}\sum_{i=1}^{K}\cos\big(a_t(s_i), a_{t-1}(s_i)\big)$ |

\* Notation: $V_t(s)$ critic prediction, $H_t$ probe-set representations, $a_t(s)$ actor mean pre-squash action.

The left two columns of Fig. 8 show the evolution of signed value shift and value churn. In simpler environments, SPPO reduces the magnitude of update-to-update value changes in the early and middle phases, while in more challenging environments, the gap to PPO is smaller and less consistent. This suggests that re-estimation of the value function remains necessary as the policy continues to evolve in complex tasks. It also indicates that overly restricting critic updates can limit the critic's ability to track the rapidly changing state distribution induced by policy improvement.

SPPO exhibits a more pronounced improvement in representation stability, as reflected by lower CKA churn in the

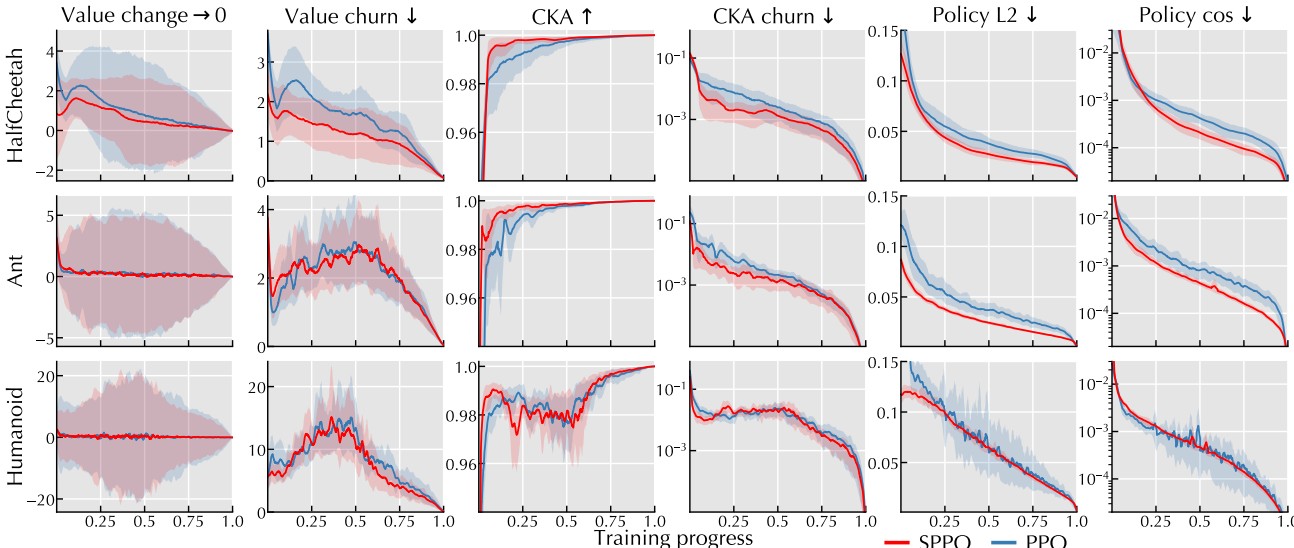

*Figure 8.* Comparison of the value change and value churn metrics between SPPO and PPO across environments.

fourth columns of Fig. 8. In `Humanoid`, SPPO maintains more stable representation geometry both in the early and in the middle-to-late stages of training. This behavior is consistent with the rapid rise of the SPPO curve in the upper-right panel of Fig. 3.

The right two columns of Fig. 8 report policy churn measured by both magnitude and directional change. SPPO yields smaller action changes and narrower variability, with the effect especially pronounced in `Humanoid`. Although the no-flip regularizer targets large directional reversals, the magnitude of action changes is also reduced. The reduction in policy churn benefits from the combined effects of the no-flip constraint and improved critic stability.

These observations align with the training curves in Fig. 3 and are consistent with the theoretical results in Sec. 3. Empirically, although the proposed method takes the form of regularization terms added to the networks, it functions to suppress oscillatory behavior in both the actor and critic over the course of training.

## 5. Conclusions

This study introduces SPPO, a drop-in latent-space stabilization scheme that preserves PPO's clipped objective and architecture. SPPO combines a CKA-based critic constraint, a no-flip regularizer on actor updates, and KDE-based advantage shaping. Theoretical analysis establishes bounds indicating reduced representation drift and fewer disruptive policy-direction changes, and biases exploration toward low-density regions. Across MuJoCo and related benchmarks, SPPO improves sample efficiency and final performance, and churn analyses indicate more stable actor and critic outputs across updates. The causal links among neural rep-

resentations, policy updates, and environment dynamics remain only partially understood; clarifying these mechanisms is an important direction for future work.

## Software and Data

The code is available at https://github.com/meiyudu0415/SPPO. The repository contains the implementation, configuration files, and scripts for reproducing the main experiments.

## Acknowledgments

The financial support from the National Natural Science Foundation of China (NSFC, Grant No. 52378182), the Shanghai Rising-Star Program (Grant No. 24YF2749100), the Collaborative Research Project under International Joint Research Laboratory of Earthquake Engineering at Tongji University and the Shanghai 2022 Science and Technology Innovation Action Plan Social Development Science and Technology Research Project (Grant No. 22dz1201700), and the Tongji Architectural Design (Group) Co., Ltd.'s Open Recruitment for Leadership Position Project: Intelligent Building Structure Design Platform (Grant No. 2024J-JB07), is gratefully acknowledged.

## Impact Statement

This paper presents work whose goal is to advance the field of Machine Learning. There are many potential societal consequences of our work, none of which we feel must be specifically highlighted here.

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

## A. Implementation Details and Full Results of CKA-Based Latent Space Analysis

This section summarizes the notation and formal definitions used in the CKA-based analysis of actor and critic latent spaces. Let the state pool collected at outer update $t$ under a fixed environment be

$$\mathcal{S}^{\star} = \{s_i\}_{i=1}^n,$$

obtained by aggregating multiple policy phases so as to cover the reachable state space. For network snapshots at updates $t$ and $t'$, the representation matrices at layer $l$ before and after activation are defined as

$$Z_t^{(l,\text{pre})} = \left[a_t^{(l)}(s_1), \ldots, a_t^{(l)}(s_n)\right]^\top \in \mathbb{R}^{n \times d_l}, \quad Z_t^{(l,\text{post})} = \left[h_t^{(l)}(s_1), \ldots, h_t^{(l)}(s_n)\right]^\top,$$

where $a_t^{(l)}$ denotes the linear output at layer $l$, $h_t^{(l)} = \sigma\big(a_t^{(l)}\big)$ is the output after applying the nonlinearity, and $d_l$ is the dimensionality of layer $l$.

Let $H = I - \frac{1}{n}\mathbf{1}\mathbf{1}^\top$ be the centering matrix. For any pair of outer updates $t, t'$ and a fixed layer $l$, the linear CKA between two representations $Z_t^{(l)}$ and $Z_{t'}^{(l)}$ (either pre- or post-activation) is defined as

$$\text{CKA}\big(Z_t^{(l)}, Z_{t'}^{(l)}\big) = \frac{\big\langle H Z_t^{(l)} Z_t^{(l)\top} H, \, H Z_{t'}^{(l)} Z_{t'}^{(l)\top} H \big\rangle_F}{\big\| H Z_t^{(l)} Z_t^{(l)\top} H \big\|_F \big\| H Z_{t'}^{(l)} Z_{t'}^{(l)\top} H \big\|_F} \in [0, 1], \tag{1}$$

where $\langle \cdot, \cdot \rangle_F$ and $\| \cdot \|_F$ denote the Frobenius inner product and norm, respectively.

Fig. 9 reports the mean trajectory with the min-max envelope of the evolution of CKA similarity in intermediate representations during training over six random seeds. Fig. 10 reports the full results for the evolution of CKA similarity in intermediate representations during training, for actor–critic networks with three hidden layers, respectively. The resulting trajectories exhibit patterns consistent with the phenomena and trends discussed in Sec. 1.

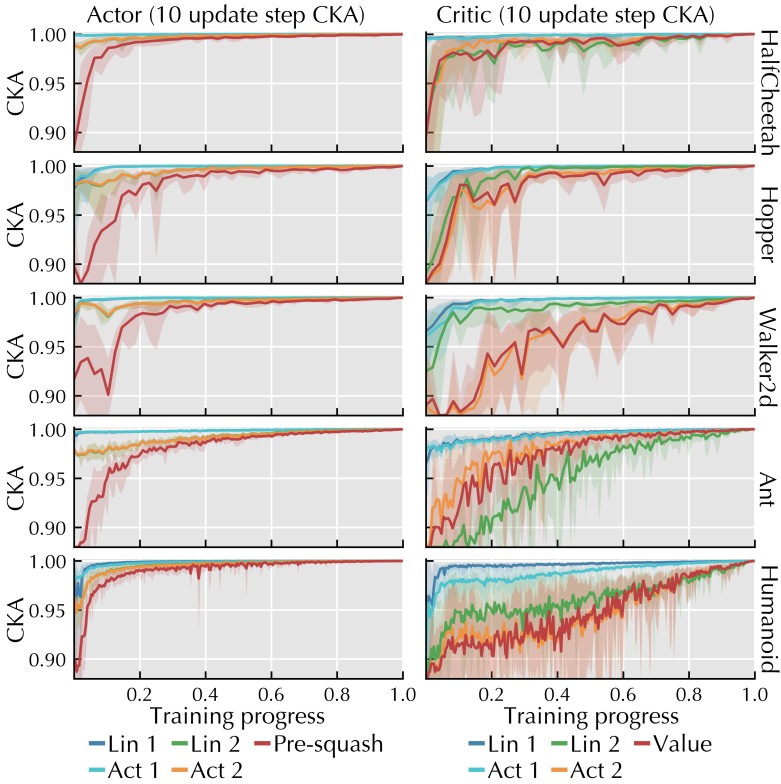

*Figure 9.* Mean trajectory across 6 random seeds with the min-max envelope.

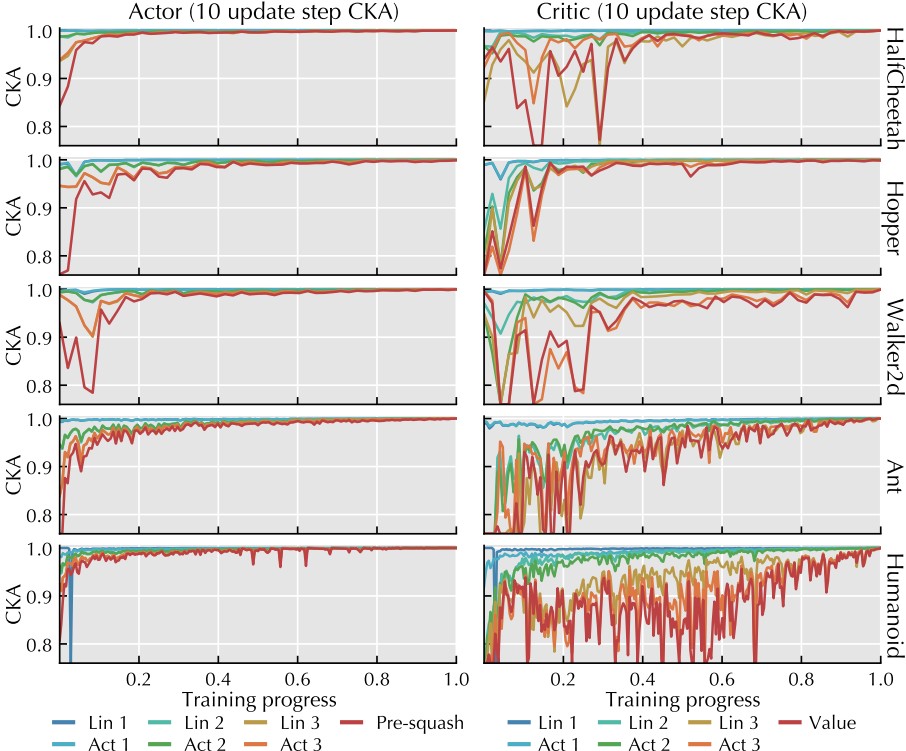

*Figure 10.* Evolution of CKA for different intermediate representations during training when the actor and critic are implemented with three hidden layers.

## B. Experimental Setup and Implementation Details

### B.1. PPO Training Hyperparameters

Tab. 5 summarizes the PPO training hyperparameters shared across all MuJoCo-v5 (Todorov et al., 2012) and DeepMind Control Suite (Tassa et al., 2018) experiments.

Unless otherwise specified, both the actor and the critic are implemented as two-layer multilayer perceptrons (excluding the output layer), with a hidden width of 128 units per layer and Tanh activations. All linear layers use orthogonal initialization (Saxe et al., 2013). Generalized advantage estimation (GAE-$\lambda$) (Schulman et al., 2015b) is employed, and advantages are normalized within each batch, as is standard practice in PPO implementations (Schulman et al., 2017). The value-function regression target is the multi-step return corresponding to TD($\lambda$)/GAE, without any additional return scaling.

The policy network outputs pre-squash means $\mu_\phi(s)$, while the diagonal standard deviation $\sigma$ is parameterized by a learnable log-standard-deviation vector that is state independent. The initial value of the learnable log-standard-deviation vector is specified per environment for more stable results and reported in Tab. 10 (MuJoCo) and Tab. 11 (DMC), respectively.

*Table 5.* Training configuration and common hyperparameters for MuJoCo-v5 and DMC experiments.

| ITEM | SYMBOL | VALUE | ITEM | VALUE |
|---|---|---|---|---|
| DISCOUNT FACTOR | $\gamma$ | 0.99 | LEARNING RATE | $3 \times 10^{-4}$ |
| GAE DECAY FACTOR | $\lambda$ | 0.95 | LR SCHEDULE | LINEAR DECAY TO 0 |
| STEPS PER ROLLOUT | $n_{\text{STEPS}}$ | 2048 | ADAM $\epsilon$ | $1 \times 10^{-5}$ |
| MINIBATCH SIZE | $B$ | 256 | GRADIENT CLIPPING THRESHOLD | 0.5 |
| EPOCHS PER UPDATE | $K$ | 8 | OBSERVATION NORMALIZATION | RUNNING MEAN/VARIANCE |
| PPO CLIPPING THRESHOLD | $\epsilon$ | 0.2 | ACTION SPACE | TANH SQUASHING |
| VALUE LOSS COEFFICIENT | $c_v$ | 0.5 | | |
| ENTROPY COEFFICIENT | $c_{\text{ENT}}$ | 0.001 | | |

## B.2. Network Architecture and Implementation Details for Pixel-Based Atari Environments

For the pixel-based Atari experiments, the ALE interface provided by `Gymnasium` (Towers et al., 2024) is used, with five discrete-action games: `BeamRider-v5`, `Breakout-v5`, `Pong-v5`, `Qbert-v5`, and `SpaceInvaders-v5`. All Atari experiments are run with vectorized environments comprising 16 parallel instances.

Observation preprocessing follows common practice in Atari PPO implementations. Color frames are converted to single-channel grayscale, images are downsampled to a resolution of $84 \times 84$, episode termination is determined by the game logic, and the last four frames are stacked along the channel dimension. After preprocessing, a single environment observation has shape $(4, 84, 84)$. Before being fed into the network, pixel values are normalized to the $[0, 1]$ range. No additional running mean/variance normalization is applied to these inputs.

The network architecture also follows standard Atari PPO practice and adopts a shared convolutional backbone for the actor–critic structure, as summarized in Tab. 6. All convolutional and fully connected layers use orthogonal initialization.

*Table 6.* Network architecture for PPO and SPPO in Atari experiments. The actor and critic share the backbone (Layers 1–5) and use separate linear heads.

| LAYER | ACTOR | CRITIC |
|---|---|---|
| 1 | CONV2D, CHANNELS $= 32$, KERNEL $= 8 \times 8$, STRIDE $= 4$, ACTIVATION = ReLU | |
| 2 | CONV2D, CHANNELS $= 64$, KERNEL $= 4 \times 4$, STRIDE $= 2$, ACTIVATION = ReLU | |
| 3 | CONV2D, CHANNELS $= 64$, KERNEL $= 3 \times 3$, STRIDE $= 1$, ACTIVATION = ReLU | |
| 4 | FLATTEN | |
| 5 | LINEAR, DIM $= 512$, ACTIVATION = ReLU | |
| 6 | LINEAR (POLICY HEAD) | LINEAR (VALUE HEAD) |

Reward clipping is used to stabilize training, while raw (unclipped) returns are reported at evaluation time, in line with common Atari PPO practice. The corresponding PPO training configurations for Atari experiments are summarized in Tab. 7.

*Table 7.* PPO training configuration and hyperparameters for Atari experiments.

| ITEM | SYMBOL | VALUE | ITEM | VALUE |
|---|---|---|---|---|
| DISCOUNT FACTOR | $\gamma$ | 0.99 | LEARNING RATE | $2.5 \times 10^{-4}$ |
| GAE DECAY FACTOR | $\lambda$ | 0.95 | LR SCHEDULE | LINEAR DECAY TO 0 |
| STEPS PER ROLLOUT (PER ENV) | $n_{\text{STEPS}}$ | 128 | ADAM $\epsilon$ | $1 \times 10^{-5}$ |
| MINIBATCH SIZE | $B$ | 256 | GRADIENT CLIPPING THRESHOLD | 0.5 |
| EPOCHS PER UPDATE | $K$ | 4 | | |
| PPO CLIPPING THRESHOLD | $\epsilon$ | 0.1 | | |
| VALUE LOSS COEFFICIENT | $c_v$ | 0.5 | | |
| ENTROPY COEFFICIENT | $c_{\text{ENT}}$ | 0.01 | | |

## B.3. Hyperparameters for CHAIN, PEER, and PFO

**CHAIN. (Tang & Berseth, 2024)** CHAIN augments standard PPO with an additional constraint on the drift of the policy's action mean. A queue of historical policies is thus maintained during training. When computing the regularization term, the second most recent policy in the queue is used as the reference policy $\pi_{\theta^{\text{ref}}}$. For states $s$ in a dedicated sub-batch $\mathcal{B}_{\text{reg}}$ used only for regularization, the L2 penalty on the mean-action drift is given by

$$L_{\text{reg}}(\theta) = \mathbb{E}_{s \in \mathcal{B}_{\text{reg}}} \left[ \| \mu_\theta(s) - \mu_{\theta^{\text{ref}}}(s) \|_2^2 \right].$$

The total loss is obtained by adding $\alpha_t L_{\text{reg}}(\theta)$ to the standard PPO loss, where the coefficient $\alpha_t$ is automatically adjusted during training based on the ratio between the average policy loss and the average regularization loss. Implementation details and further analysis can be found in (Tang & Berseth, 2024).

**PEER. (He et al., 2023)** Relative to PPO, PEER adds a regularization term on consecutive critic representations. PEER constructs an auxiliary loss from the critic's latent representations $h(s)$ and $h(s')$ of consecutive states $(s, s')$. Both are first

*Table 8.* CHAIN hyperparameters.

| DESCRIPTION | SYMBOL | VALUE |
|---|---|---|
| INITIAL REGULARIZATION COEFFICIENT | $\alpha_0$ | 100.0 |
| TARGET RELATIVE LOSS RATIO | $\kappa$ | 0.1 |
| MAXIMUM NUMBER OF HISTORICAL POLICIES | $N_{\mathrm{hist}}$ | 10 |
| WARM-UP ITERATIONS BEFORE ADAPTATION | $N_{\mathrm{warm}}$ | 50 |
| RUNNING-AVERAGE WINDOW LENGTH | $N_{\mathrm{avg}}$ | 100 |

L2-normalized along the feature dimension to obtain $\tilde{h}(s)$ and $\tilde{h}(s')$, and the cosine similarity is used as the loss:

$$L_{\mathrm{PEER}} = \mathbb{E}_{(s,s')}\big[\tilde{h}(s)^{\top}\tilde{h}(s')\big].$$

Here, $h(s)$ is obtained via a forward pass through the current critic and participates in backpropagation, while $h(s')$ is treated as a constant (no gradient). The term $L_{\mathrm{PEER}}$ is added to the standard PPO loss with a fixed weight $\beta$, which is set to $5 \times 10^{-4}$ in this study. The full implementation follows (He et al., 2023).

**PFO. (Moalla et al., 2024)** PFO applies feature regularization in the actor's pre-activation space. During sampling, the pre-activation vector $z_{\mathrm{old}}(s)$ from the last linear layer of the policy network is recorded and stored in the rollout buffer together with the trajectory. In subsequent PPO updates, the updated policy is used to recompute $z_{\mathrm{cur}}(s)$, and a feature-preservation loss is constructed:

$$L_{\mathrm{PFO}} = \mathbb{E}_s\big[\|z_{\mathrm{cur}}(s) - z_{\mathrm{old}}(s)\|_2^2\big].$$

The total loss is obtained by adding $\lambda L_{\mathrm{PFO}}$ to the standard PPO loss, with $\lambda$ set to $1.0$ in this study. The implementation details follow (Moalla et al., 2024).

### B.4. Default Hyperparameters for SPPO

SPPO introduces nine additional hyperparameters. Their default values correspond to the *default* setting in Tab. 9. $\lambda_{\mathrm{CKA}}$, $\lambda_{\mathrm{NF}}$, and $\lambda_{\mathrm{KDE}}$ control the weights of the three components in the loss function or advantage shaping, respectively. $L_{\mathrm{seq}}$, $N_{\mathrm{seq}}$, and $B_{\mathrm{CKA}}$ denote, respectively, the short-sequence length, the number of sequences, and the number of interaction rounds covered by the history buffer when computing the CKA term (for example, if each update uses 2048 interaction steps, then $B_{\mathrm{CKA}} = 1$ corresponds to caching the most recent 2048 samples). $N_{\mathrm{NF}}$ and $B_{\mathrm{NF}}$ are the sample count and number of historical rounds used in the no-flip computation, analogously to $B_{\mathrm{CKA}}$ but on the actor side. And $p$ is the quantile used as the KDE novelty baseline. Tab. 9 also lists the parameter combinations used in the sensitivity analysis reported in Appendix C.6.

*Table 9.* Hyperparameter configurations for SPPO. The *default* row indicates the default setting used in the main experiments. The groups $\{C_i\}$, $\{N_i\}$, and $\{K_i\}$ are used in the sensitivity analysis in Appendix C.6.

| ID | $\lambda_{\mathrm{CKA}}$ | $L_{\mathrm{seq}}$ | $N_{\mathrm{seq}}$ | $B_{\mathrm{CKA}}$ | ID | $\lambda_{\mathrm{NF}}$ | $N_{\mathrm{NF}}$ | $B_{\mathrm{NF}}$ | ID | $\lambda_{\mathrm{KDE}}$ | $p$ |
|---|---|---|---|---|---|---|---|---|---|---|---|
| DEFAULT | 10 | 4 | 256 | 5 | DEFAULT | 1 | 1024 | 3 | DEFAULT | 1 | 0.2 |
| C1 | 10 | 4 | 128 | 5 | N1 | 1 | 512 | 3 | K1 | 1 | 0.1 |
| C2 | 10 | 4 | 64 | 5 | N2 | 1 | 256 | 3 | K2 | 1 | 0.5 |
| C3 | 10 | 4 | 32 | 5 | N3 | 1 | 128 | 3 | K3 | 0.5 | 0.2 |
| C4 | 10 | 16 | 64 | 5 | N4 | 0.5 | 1024 | 3 | K4 | 0.1 | 0.2 |
| C5 | 10 | 32 | 32 | 5 | N5 | 0.1 | 1024 | 3 | K5 | 2 | 0.2 |
| C6 | 10 | 64 | 16 | 5 | N6 | 1 | 1024 | 1 | | | |
| C7 | 5 | 4 | 256 | 5 | N7 | 1 | 1024 | 5 | | | |
| C8 | 1 | 4 | 256 | 5 | | | | | | | |
| C9 | 10 | 4 | 256 | 1 | | | | | | | |
| C10 | 10 | 4 | 256 | 10 | | | | | | | |

## B.5. Environment Configurations and Training Steps

The experiments cover environments from MuJoCo (Todorov et al., 2012), DMC (Tassa et al., 2018), and Atari 2600 games from ALE (Bellemare et al., 2013). For each environment, the total number of training steps and the initial policy log-standard-deviation are set individually, but are held fixed across all algorithms within that environment for fair comparison. The initial log-standard-deviation is selected based on preliminary tuning runs that calibrate the baseline PPO to achieve stable training behavior. All MuJoCo and DMC tasks use their v5 environment versions. Tabs. 10, 11, and 12 summarize the corresponding configurations.

*Table 10.* MuJoCo environments configurations.

| ENVIRONMENT | HOPPER-V5 | WALKER2D-V5 | HALFCHEETAH-V5 | ANT-V5 | HUMANOID-V5 |
|---|---|---|---|---|---|
| TOTAL STEPS | 1M | 1M | 1M | 3M | 5M |
| INITIAL $\log \sigma$ | $-0.5$ | $-0.5$ | $-0.5$ | $-1.0$ | $-1.0$ |

*Table 11.* DeepMind Control Suite environment configurations.

| ENVIRONMENT | CHEETAH RUN | WALKER WALK | WALKER RUN | QUADRUPED WALK | HUMANOID WALK |
|---|---|---|---|---|---|
| TOTAL STEPS | 1M | 1M | 1M | 3M | 5M |
| INITIAL $\log \sigma$ | $-0.5$ | $-0.5$ | $-0.5$ | $-0.5$ | $-1.0$ |

*Table 12.* ALE environment configurations for Atari 2600 games.

| ENVIRONMENT | PONG | BREAKOUT | BEAMRIDER | QBERT | SPACEINVADERS |
|---|---|---|---|---|---|
| TOTAL STEPS | 5M | 5M | 5M | 5M | 5M |

## B.6. Software and Hardware Setup

The implementation is based on Python 3.11.14, PyTorch 2.6.0, Gymnasium 1.2.0, and MuJoCo 3.3.6. All experiments are run on a workstation equipped with an AMD EPYC 9474F CPU. Due to the relatively small model sizes, CPU training is more efficient than GPU training in this setting. For example, on `Humanoid-v5`, training a single SPPO seed for 5M steps takes approximately 3 hours, while PPO training for the same number of steps takes about 2 hours; the difference in wall-clock time is primarily due to the KDE computation required for exploration-driven advantage shaping. All methods and hyperparameter configurations are run independently on the same hardware without special system-level optimizations.

# C. Extra Results and Analysis

## C.1. Final Evaluation Returns for the Experiments

Tabs. 13–16 report the mean and standard deviation of the final evaluation return for the MuJoCo experiments under different network capacities, complementing the main results in Sec. 4. These statistics are computed by aggregating evaluation returns from the last ten training updates across six random seeds.

*Table 13.* Final evaluation returns for MuJoCo tasks (3×128).

| ALGO | HOPPER | WALKER2D | HALFCHEETAH | ANT | HUMANOID |
|---|---|---|---|---|---|
| SPPO | $2352.51 \pm 943.40$ | $\mathbf{4016.55 \pm 1085.07}$ | $\mathbf{5741.28 \pm 643.73}$ | $4221.87 \pm 855.68$ | $5731.12 \pm 2398.38$ |
| CHAIN | $918.79 \pm 415.70$ | $1216.02 \pm 536.87$ | $3665.91 \pm 377.60$ | $3705.77 \pm 1298.63$ | $\mathbf{6124.22 \pm 236.16}$ |
| PEER | $1560.39 \pm 820.11$ | $3034.62 \pm 1187.72$ | $5522.31 \pm 998.73$ | $\underline{4777.77 \pm 496.21}$ | $5229.80 \pm 1912.90$ |
| PFO | $\mathbf{2646.42 \pm 930.31}$ | $3671.02 \pm 678.29$ | $4791.10 \pm 1173.32$ | $\mathbf{4987.43 \pm 610.83}$ | $4174.00 \pm 2406.95$ |
| PPO | $1738.57 \pm 881.77$ | $\underline{3738.59 \pm 771.23}$ | $\underline{5642.86 \pm 464.61}$ | $3800.40 \pm 1254.36$ | $4312.24 \pm 2327.98$ |

*Table 14.* Final evaluation returns for MuJoCo tasks (4×128).

| ALGO | HOPPER | WALKER2D | HALFCHEETAH | ANT | HUMANOID |
|------|--------|----------|-------------|-----|----------|
| SPPO | **2662.95 ± 862.53** | **4158.13 ± 862.15** | **5359.86 ± 576.33** | 3172.59 ± 1247.87 | 5318.93 ± 1759.82 |
| CHAIN | 890.30 ± 383.82 | 1294.46 ± 307.00 | 3276.34 ± 868.18 | 4023.27 ± 1160.77 | 5307.40 ± 1751.27 |
| PEER | 1743.30 ± 806.80 | 3505.91 ± 1108.25 | 4849.29 ± 1204.53 | 4303.71 ± 1597.63 | **5420.24 ± 1780.30** |
| PFO | 1746.19 ± 841.35 | 3306.60 ± 1099.32 | 5016.70 ± 829.03 | **4350.81 ± 1585.09** | 3363.97 ± 2580.05 |
| PPO | 2346.88 ± 942.38 | 3046.19 ± 1311.57 | 4452.85 ± 1031.37 | 2441.30 ± 1486.14 | 4394.74 ± 1620.04 |

*Table 15.* Final evaluation returns for MuJoCo tasks (2×256).

| ALGO | HOPPER | WALKER2D | HALFCHEETAH | ANT | HUMANOID |
|------|--------|----------|-------------|-----|----------|
| SPPO | 2647.57 ± 845.62 | 3699.49 ± 1299.23 | **6132.63 ± 870.57** | 4537.68 ± 1574.98 | **7522.72 ± 889.67** |
| CHAIN | 1929.62 ± 1076.99 | 2220.73 ± 944.34 | 4063.67 ± 849.45 | 4656.99 ± 712.73 | 6605.54 ± 331.11 |
| PEER | 2576.17 ± 904.98 | 3739.76 ± 1098.71 | 4715.36 ± 1648.05 | **4841.26 ± 920.15** | 4354.98 ± 2774.39 |
| PFO | **2873.44 ± 897.72** | **3774.05 ± 1216.95** | 5386.21 ± 895.97 | 4632.88 ± 1115.64 | 4477.92 ± 2295.95 |
| PPO | 2378.97 ± 885.25 | 3164.08 ± 1596.92 | 4397.09 ± 1948.60 | 3736.20 ± 1122.98 | 3760.92 ± 2164.18 |

*Table 16.* Final evaluation returns for MuJoCo tasks (2×512).

| ALGO | HOPPER | WALKER2D | HALFCHEETAH | ANT | HUMANOID |
|------|--------|----------|-------------|-----|----------|
| SPPO | 3119.05 ± 798.91 | 4676.54 ± 331.19 | **6568.26 ± 1204.32** | 4302.58 ± 1311.15 | 5305.42 ± 2328.66 |
| CHAIN | 3082.30 ± 782.05 | 3742.99 ± 720.59 | 4440.46 ± 1825.38 | **4476.65 ± 1267.59** | **5533.42 ± 2111.65** |
| PEER | 3259.37 ± 685.83 | **4780.80 ± 948.95** | 4942.14 ± 1817.46 | 4197.12 ± 953.20 | 1946.78 ± 1776.72 |
| PFO | **3345.86 ± 598.93** | 4132.68 ± 759.52 | 4562.46 ± 1623.21 | 4109.65 ± 842.83 | 2887.65 ± 1995.77 |
| PPO | 2826.98 ± 910.11 | 4521.96 ± 382.68 | 5383.35 ± 590.34 | 2993.72 ± 1281.25 | 3829.06 ± 2057.32 |

## C.2. RLiable Metrics under Different Network Architectures

Fig. 11 presents the evolution of RLiable metrics during training for each algorithm under different network capacities on MuJoCo environments. And the performance profiles of each algorithm across different network capacities under the RLiable evaluation protocol are shown in Fig. 12.

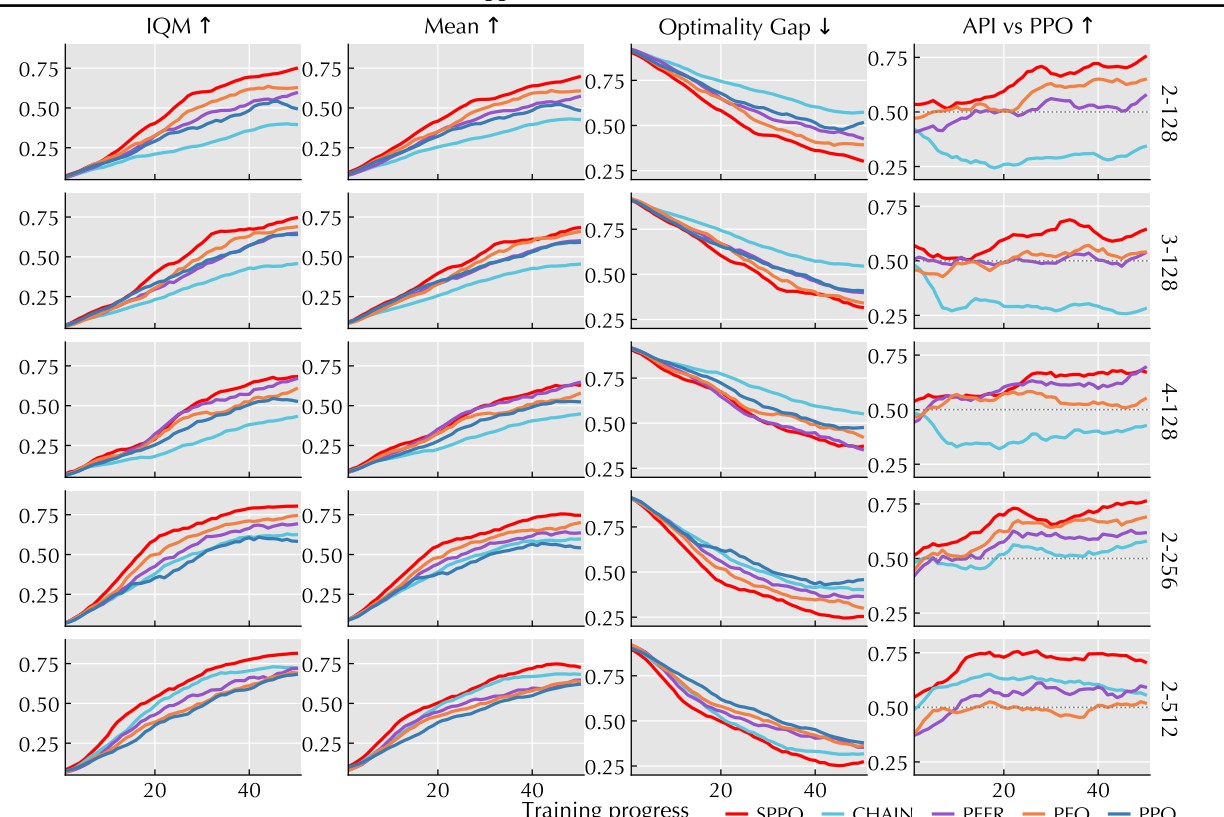

*Figure 11.* Evolution of RLiable metrics during training for different algorithms and network capacities on MuJoCo environments.

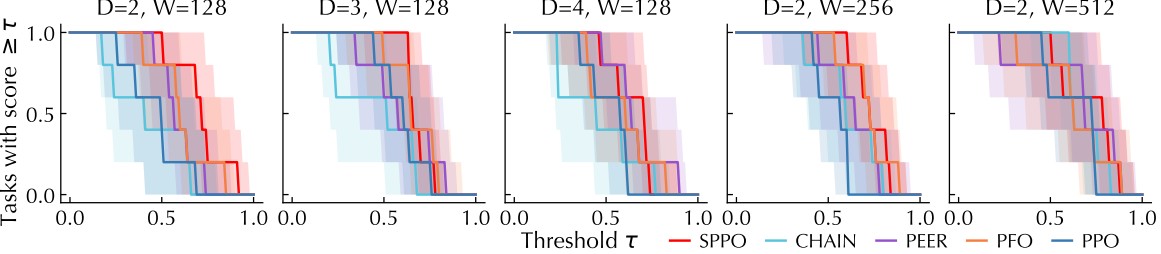

*Figure 12.* Performance profiles of the five algorithms on MuJoCo-v5.

Fig. 13 reports the final RLiable metrics for the same set of configurations. Across network depths and widths, SPPO consistently exhibits advantages in both sample efficiency and final performance. All algorithms benefit from increased model capacity. In terms of depth and width, CHAIN shows a pronounced performance improvement when the network is widened. Compared with other improved algorithms and vanilla PPO, SPPO maintains relatively narrow 95% bootstrap confidence intervals across configurations, indicating more stable training dynamics and reduced variability across random seeds.

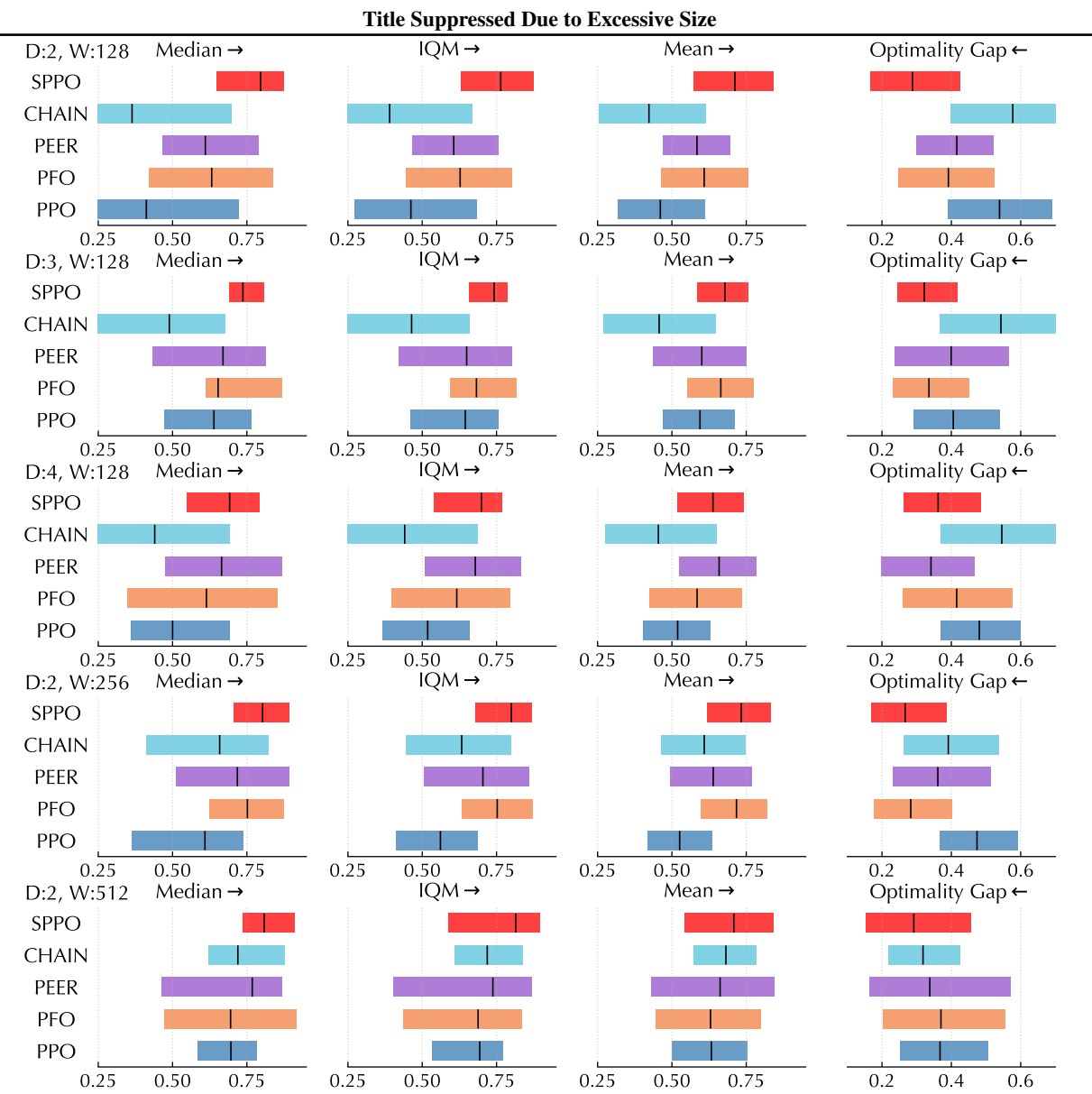

*Figure 13.* Final RLiable metrics for different algorithms and network capacities on MuJoCo environments.

## C.3. Full Training Curves for the Ablation Study

Fig. 14 shows the full training-curve comparisons for the SPPO ablation study on MuJoCo environments, corresponding to the component-wise on/off configurations analyzed in Sec. 4.2.

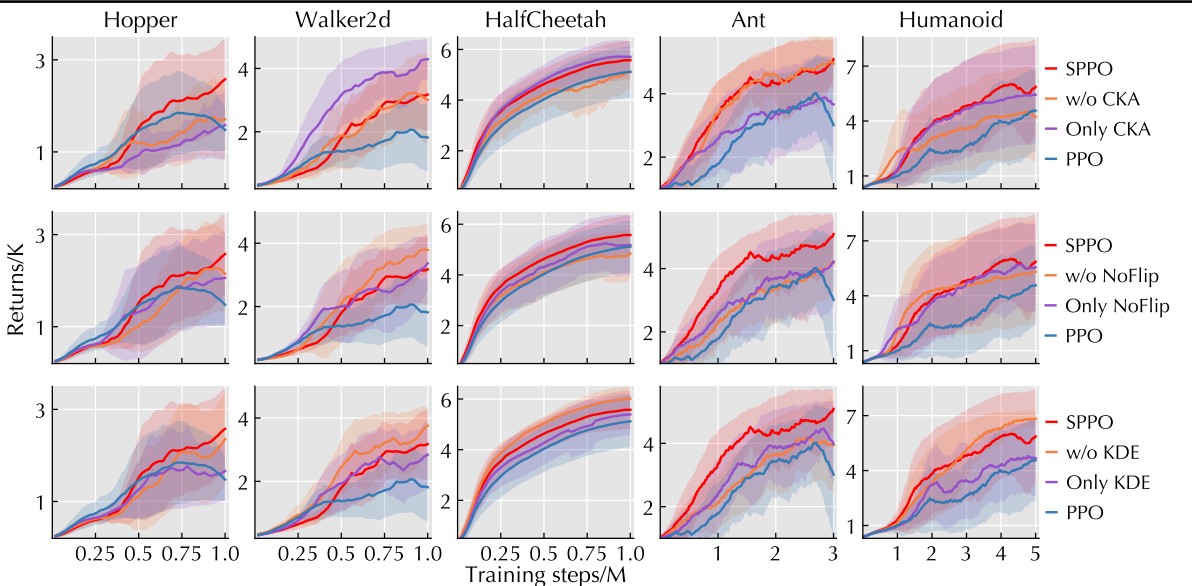

*Figure 14.* Full training-curve comparisons for the SPPO ablation study on MuJoCo environments.

## C.4. Formulation of Churn Metrics

Tab. 4 summarizes the six metrics that are constructed to quantify the changes between two consecutive updates as discussed in Sec. 4.3. In Tab. 4, $t = 1, \ldots, T - 1$ denotes the index of the policy-update step, and each metric $\text{Metric}_t$ characterizes the change between update $t - 1$ and update $t$. Let $\{s_1, \ldots, s_K\}$ be the fixed set of $K$ representative states sampled from the reachable state distribution (with $K = 2048$ in the experiments). For any state $s$, $V_t(s)$ denotes the scalar value estimate produced by the critic at checkpoint $t$, and $a_t(s) \in \mathbb{R}^{d_a}$ denotes the continuous action output by the actor at checkpoint $t$. The matrix $H_t \in \mathbb{R}^{K \times d_h}$ denotes the critic's post-activation latent representations at checkpoint $t$ on these $K$ states, with each row corresponding to the feature vector of one state.

The notation $\| \cdot \|_2$ and $\| \cdot \|_F$ denote the Euclidean norm and Frobenius norm, respectively, and $\langle \cdot, \cdot \rangle$ denotes the inner product between vectors. The linear CKA similarity $\text{CKA}(H_t, H_{t-1})$ is defined via Gram matrices as

$$\text{CKA}(X, Y) = \frac{\|Y^\top X\|_F^2}{\|X^\top X\|_F \, \|Y^\top Y\|_F},$$

and is used to quantify the consistency between intermediate representations obtained at two consecutive updates on the same set of probe states.

## C.5. Additional Results for the Churn Evaluation

For brevity, the main text in Sec. 4.3 focuses on the churn metrics for the more challenging MuJoCo tasks (HalfCheetah-v5, Ant-v5 and Humanoid-v5). Fig. 15 complement the results by reporting the same set of value-, representation-, and policy-level churn metrics on the simpler Hopper-v5 and Walker2d-v5 environments. In these simpler environments, SPPO exhibits a more pronounced stabilizing effect relative to PPO. Inter-update changes in value predictions and policy outputs are reduced, and the critic representations maintain higher CKA similarity and lower CKA churn throughout training.

## C.6. Hyperparameter Sensitivity of SPPO

This section presents a grid-search sensitivity analysis over the hyperparameter combinations in Tab. 9, with results shown in Fig. 16. Each row in Fig. 16 corresponds to one hyperparameter group. The top four rows vary the CKA regularization settings (C1–C10), the middle three rows vary the no-flip directional-consistency settings (N1–N7), and the bottom two rows vary the KDE-based novelty-shaping settings (K1–K5). The dark blue curve (BL) denotes the default configuration, whereas the remaining curves vary one hyperparameter at a time while keeping all others fixed at their default values. Definitions of all hyperparameters are provided in Appendix B.4.

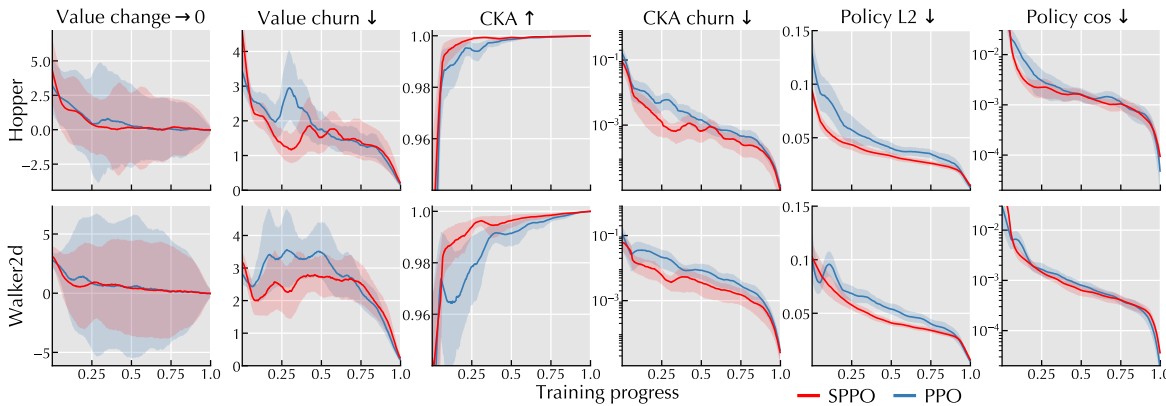

*Figure 15.* Six churn metrics for PPO and SPPO on Hopper-v5 and Walker2d-v5.

The default SPPO configuration, selected based on preliminary tuning, falls within a favorable region for most environments. In Fig. 16, the dark blue curve is above most solid curves in the majority of panels and is typically higher than the red dashed curve corresponding to vanilla PPO. Two deviations are observed. In Hopper, performance is more sensitive to the hyperparameters and SPPO exceeds PPO only under a subset of settings. In Walker2d, slightly weaker regularization than the default can yield marginally better learning curves.

Across environments, the preferred regularization strength varies. In Walker2d, smaller values of $\lambda_{\mathrm{CKA}}$, $\lambda_{\mathrm{NF}}$, and $\lambda_{\mathrm{KDE}}$ tend to improve returns. In higher-dimensional and more dynamically complex tasks such as Ant and Humanoid, moderately larger weights (matching the default setting in Fig. 16) more effectively damp oscillations and improve final performance.

The historical buffer lengths exhibit a characteristic U-shaped trend. When $B_{\mathrm{CKA}}$ or $B_{\mathrm{NF}}$ is too small, the regularizers depend almost entirely on the most recent update and provide limited smoothing of per-update noise. When the buffers are too long, the regularizers incorporate a larger fraction of off-policy data, which slows adaptation to the current policy. The default choices $B_{\mathrm{CKA}} = 5$ and $B_{\mathrm{NF}} = 3$ provide stable behavior across the five environments, whereas extreme values (e.g., 1 or 10) lead to clear degradation on Ant and Humanoid.

The quantile threshold $p$ should remain small, as a high threshold (e.g., $p = 0.5$) can over-distort advantages, lower threshold ($p = 0.1$) behaves similarly to the default ($p = 0.2$), suggesting that restricting shaping to the sparsest states is more robust.

These results suggest that the baseline configuration in Tab. 9 is close to the center of the stable regimes for the examined hyperparameters, providing a practical trade-off between performance and robustness across environments. It should also be noted that these 9 hyperparameters do not have equal tuning priority. The three weight parameters are the main control variables, while the remaining hyperparameters can generally be kept at their default values. The main purpose of Fig. 16 is to show how each parameter affects training and final return, thereby clarifying the sensitivity of SPPO. It is agreed that joint sensitivity under multi-parameter tuning deserves deeper analysis. However, the current sensitivity analysis is mainly intended to demonstrate the usability of the default configuration and local robustness, rather than to fully characterize interactions in a high-dimensional hyperparameter space.

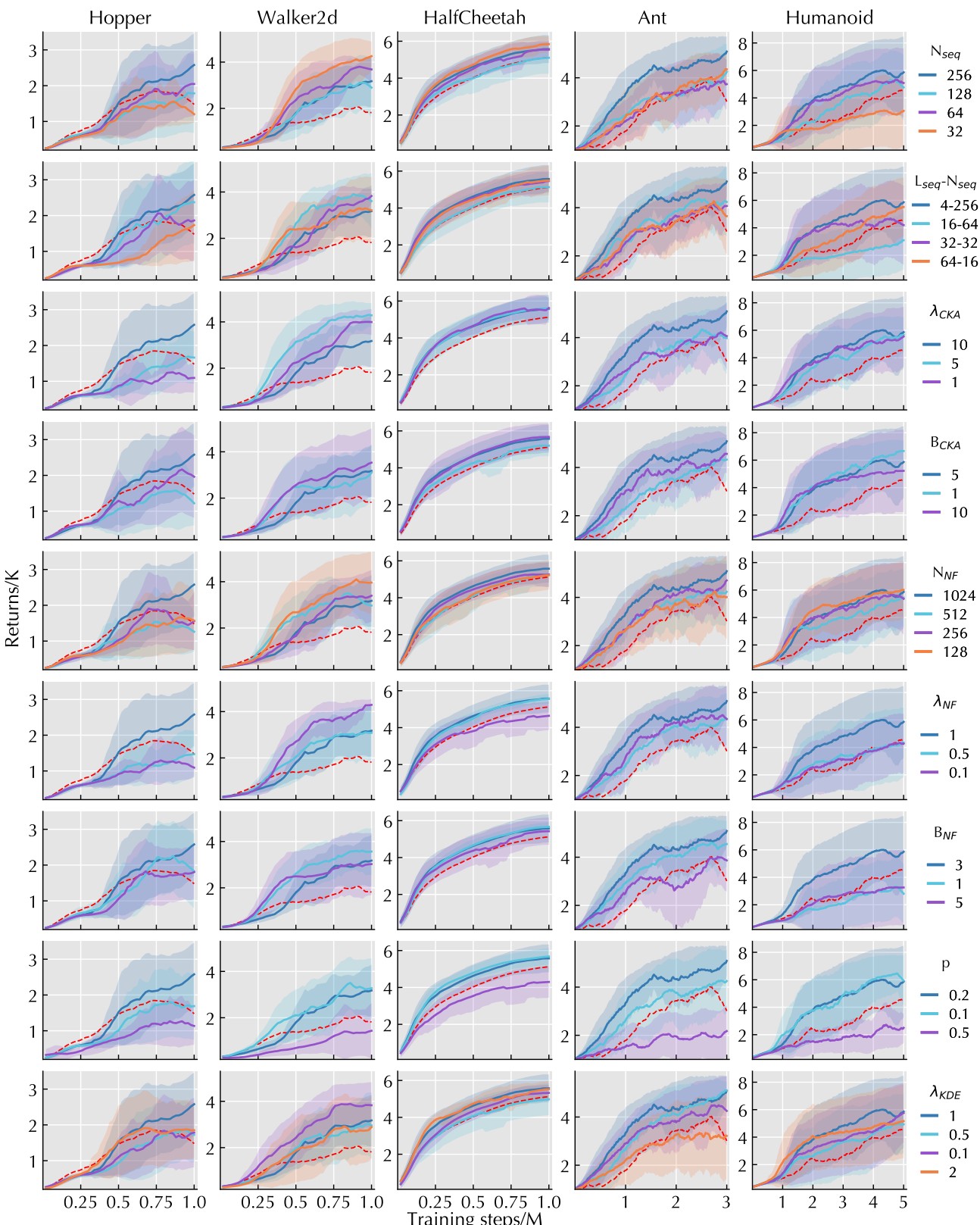

*Figure 16.* Hyperparameter sensitivity of SPPO.

# D. Proofs and Detailed Derivations

Unless otherwise stated, $\|\cdot\|$ denotes the spectral norm and $\|\cdot\|_F$ denotes the Frobenius norm. The Löwner order $A \succeq B$ means that $A - B$ is positive semidefinite. All derivations are carried out under a single small update step with first-order linearization of the encoder, and external normalizations (such as running normalization) are treated as frozen within the current update.

## D.1. CKA Regularization: Quadratic Approximation and Tighter Upper Bounds

This section provides detailed derivations for Proposition 3.1–Corollary 3.3 in Sec. 3.2.

### D.1.1. CRITIC UPDATE UNDER LINEARIZATION AND RIDGE-REGRESSION FORM

For a single small update step, fixing $\omega$ and linearizing the encoder output $h_\theta(s)$ gives

$$h_{\theta+\Delta\theta}(s) \approx h_\theta(s) + J_\theta(s)\,\Delta\theta,$$

where $J_\theta(s) = \partial h_\theta(s)/\partial\theta$ is the Jacobian of the encoder at $s$. Under this first-order approximation, the updated value estimate on a sample $s_i$ can be written as

$$\hat{V}_{\theta+\Delta\theta,\omega}(s_i) \approx \hat{V}_{\theta,\omega}(s_i) + \omega^\top J_\theta(s_i)\,\Delta\theta.$$

Define the design matrix $X \in \mathbb{R}^{n \times d}$ with the $i$-th row

$$X_i = \left(J_\theta(s_i)^\top \omega\right)^\top,$$

and let $r \in \mathbb{R}^n$ denote the residual vector. The quadratic approximation of the objective over a batch of $n$ samples is

$$\min_{\Delta\theta} \sum_{i=1}^n \left(\hat{V}_{\theta+\Delta\theta,\omega}(s_i) - y_i\right)^2 + \lambda\|\Delta\theta\|^2 \;\approx\; \min_{\Delta\theta} \|X\Delta\theta - r\|^2 + \lambda\|\Delta\theta\|^2,$$

where $\lambda > 0$ is an $\ell_2$ regularization coefficient on the encoder parameters. This is a ridge-regression problem with closed-form solution

$$\Delta\theta^* = (X^\top X + \lambda I)^{-1} X^\top r.$$

For convenience, define

$$A_0 \;=\; X^\top X + \lambda I.$$

### D.1.2. QUADRATIC UPPER BOUND FOR SHORT-SEQUENCE CKA AND THE MATRIX $M_{\mathrm{CKA}}$

Consider a short sequence of length $L$,

$$\mathcal{S}_s = \{s_t, \ldots, s_{t+L-1}\}.$$

Passing this sequence through the encoder before and after the update yields latent matrices $H_{\mathrm{old}}, H_{\mathrm{new}} \in \mathbb{R}^{L \times H}$, where the $k$-th row corresponds to $h_\theta(s_{t+k-1})^\top$ and $h_{\theta+\Delta\theta}(s_{t+k-1})^\top$, respectively. Let $C_L = I_L - \frac{1}{L}\mathbf{1}\mathbf{1}^\top$ be the centering matrix along the sample dimension. The centered representations are

$$\tilde{H}_{\mathrm{old}} = C_L H_{\mathrm{old}}, \qquad \tilde{H}_{\mathrm{new}} = C_L H_{\mathrm{new}}.$$

The corresponding (unnormalized) Gram matrices are

$$B = \tilde{H}_{\mathrm{old}}^\top \tilde{H}_{\mathrm{old}}, \qquad A = \tilde{H}_{\mathrm{new}}^\top \tilde{H}_{\mathrm{new}}.$$

Define their normalized versions

$$\hat{B} = \frac{B}{\|B\|_F}, \qquad \hat{A} = \frac{A}{\|A\|_F}.$$

The linear CKA similarity can then be written as

$$\mathrm{CKA}(\tilde{H}_{\mathrm{new}}, \tilde{H}_{\mathrm{old}}) = \frac{\langle A, B\rangle_F}{\|A\|_F\,\|B\|_F} = 1 - \frac{1}{2}\|\hat{A} - \hat{B}\|_F^2.$$

Thus the structure-preserving term becomes

$$L_{\text{struct}} = \lambda_{\text{CKA}}(1 - \text{CKA}) = \frac{\lambda_{\text{CKA}}}{2} \|\hat{A} - \hat{B}\|_F^2.$$

Under a small encoder update with first-order linearization,

$$H_{\text{new}} \approx H_{\text{old}} + J_s \Delta\theta,$$

where $J_s$ is the Jacobian stacked over the short sequence $\mathcal{S}_s$ (of shape $L \times d$). Hence

$$\tilde{H}_{\text{new}} = C_L H_{\text{new}} \approx C_L H_{\text{old}} + C_L J_s \Delta\theta = \tilde{H}_{\text{old}} + C_L J_s \Delta\theta.$$

By performing a second-order Taylor expansion of $\hat{A}$ around $\Delta\theta = 0$ and using the Lipschitz continuity of $\|\cdot\|_F$ with respect to its argument, there exists a positive semidefinite matrix $G \succeq 0$, depending only on the old sequence, such that within a sufficiently small neighborhood of $\Delta\theta = 0$,

$$L_{\text{struct}} = \frac{\lambda_{\text{CKA}}}{2} \|\hat{A} - \hat{B}\|_F^2 \ \leq \ 8\lambda_{\text{CKA}} \, \Delta\theta^\top J_s^\top G J_s \, \Delta\theta.$$

Define

$$M_{\text{CKA}} \ := \ 8\lambda_{\text{CKA}} \, J_s^\top G J_s \succeq 0.$$

Under the first-order linearization and this quadratic upper bound, introducing the CKA structure term is equivalent to adding a quadratic regularizer in $\Delta\theta$,

$$L_{\text{struct}} \ \lesssim \ \Delta\theta^\top M_{\text{CKA}} \, \Delta\theta.$$

Consequently, the approximate optimization problem with CKA regularization can be written as

$$\min_{\Delta\theta} \|X\Delta\theta - r\|^2 + \lambda\|\Delta\theta\|^2 + \Delta\theta^\top M_{\text{CKA}} \, \Delta\theta,$$

whose closed-form solution is

$$\Delta\theta^* = (X^\top X + \lambda I + M_{\text{CKA}})^{-1} X^\top r.$$

Define

$$A_{\text{CKA}} = X^\top X + \lambda I + M_{\text{CKA}}.$$

Clearly $A_{\text{CKA}} \succeq A_0$, and hence, in the Löwner order, $\|A_{\text{CKA}}^{-1}\| \leq \|A_0^{-1}\|$.

### D.1.3. POINTWISE REPRESENTATION DRIFT AND MEAN-SQUARE DRIFT OVER UNVISITED STATES

For any state $s'$, the change in its latent representation under the update satisfies

$$\delta h_\theta(s') \approx J_\theta(s') \, \Delta\theta^*,$$

and by submultiplicativity,

$$\|\delta h_\theta(s')\| \ \leq \ \|J_\theta(s')\| \, \|A_{\text{CKA}}^{-1} X^\top\| \, \|r\|.$$

Since $A_{\text{CKA}} \succeq A_0$ implies $\|A_{\text{CKA}}^{-1}\| \leq \|A_0^{-1}\|$, this further yields

$$\|\delta h_\theta(s')\| \ \leq \ \|J_\theta(s')\| \, \|A_0^{-1} X^\top\| \, \|r\|.$$

This is precisely the upper bound stated in Proposition 5.1 for the pointwise drift of the latent representation.

Now consider a set of unvisited states $U = \{s'_j\}_{j=1}^m$. Stack their Jacobians row-wise as

$$J_U = \begin{bmatrix} J_\theta(s'_1) \\ \vdots \\ J_\theta(s'_m) \end{bmatrix},$$

and stack the corresponding latent changes as

$$\Delta H_U = J_U \, \Delta\theta^* = J_U A_{\mathrm{CKA}}^{-1} X^\top r.$$

Then

$$\|\Delta H_U\|_F^2 = \mathrm{Tr}(\Delta H_U^\top \Delta H_U) = r^\top X A_{\mathrm{CKA}}^{-1} J_U^\top J_U A_{\mathrm{CKA}}^{-1} X^\top r.$$

Let $\mathrm{Cov}(r)$ denote the covariance of the residuals $r$, and assume $\mathbb{E}[r] = 0$. Define

$$M = X A_{\mathrm{CKA}}^{-1} J_U^\top J_U A_{\mathrm{CKA}}^{-1} X^\top \succeq 0.$$

Then

$$\mathbb{E}\big[\|\Delta H_U\|_F^2\big] = \mathbb{E}[r^\top M r] = \mathrm{Tr}(M \, \mathrm{Cov}(r)) \leq \lambda_{\max}(\mathrm{Cov}(r)) \, \mathrm{Tr}(M),$$

where the last inequality uses

$$\mathrm{Tr}(M \, \mathrm{Cov}(r)) \leq \lambda_{\max}(\mathrm{Cov}(r)) \, \mathrm{Tr}(M).$$

On the other hand,

$$\mathrm{Tr}(M) = \mathrm{Tr}\big(J_U A_{\mathrm{CKA}}^{-1} X^\top X A_{\mathrm{CKA}}^{-1} J_U^\top\big),$$

so

$$\mathbb{E}\big[\|\Delta H_U\|_F^2\big] \leq \lambda_{\max}(\mathrm{Cov}(r)) \, \mathrm{Tr}\big(J_U A_{\mathrm{CKA}}^{-1} X^\top X A_{\mathrm{CKA}}^{-1} J_U^\top\big).$$

Using $A_{\mathrm{CKA}} \succeq A_0 \Rightarrow A_{\mathrm{CKA}}^{-1} \preceq A_0^{-1}$ in the Löwner order, one further obtains

$$\mathbb{E}\big[\|\Delta H_U\|_F^2\big] \leq \lambda_{\max}(\mathrm{Cov}(r)) \, \mathrm{Tr}\big(J_U A_0^{-1} X^\top X A_0^{-1} J_U^\top\big).$$

Thus the mean-square upper bound on the latent drift over the unvisited set $U$ becomes smaller or equal after adding the CKA regularizer, corresponding to Proposition 3.2 in Sec. 3.2.

### D.1.4. PARAMETER PERTURBATIONS ACROSS TWO BATCHES AND UPPER BOUNDS ON BOOTSTRAPPING ERROR

Consider a single update step evaluated on two (possibly different) batches or distributions, with matrices and residuals $(X, r)$ and $(X', r')$, respectively. The corresponding parameter updates are

$$\Delta\theta = A_{\mathrm{CKA}}^{-1} X^\top r, \qquad \Delta\theta' = A_{\mathrm{CKA}}'^{-1} X'^\top r',$$

where $A_{\mathrm{CKA}}'$ denotes the counterpart of $A_{\mathrm{CKA}}$ for $(X', r')$. Their difference can be decomposed as

$$\begin{aligned} \Delta\theta' - \Delta\theta = A_{\mathrm{CKA}}'^{-1} X'^\top (r' - r) + A_{\mathrm{CKA}}'^{-1} (X'^\top - X^\top) r \\ + \big(A_{\mathrm{CKA}}'^{-1} - A_{\mathrm{CKA}}^{-1}\big) X^\top r. \end{aligned}$$

For the last term, the matrix perturbation identity

$$(A + \Delta)^{-1} - A^{-1} = -A^{-1} \Delta (A + \Delta)^{-1}$$

is used. Setting $A = A_{\mathrm{CKA}}$ and $\Delta = A_{\mathrm{CKA}}' - A_{\mathrm{CKA}}$ yields

$$A_{\mathrm{CKA}}'^{-1} - A_{\mathrm{CKA}}^{-1} = -A_{\mathrm{CKA}}^{-1} (A_{\mathrm{CKA}}' - A_{\mathrm{CKA}}) A_{\mathrm{CKA}}'^{-1}.$$

Hence

$$\begin{aligned} \|\Delta\theta' - \Delta\theta\| \leq \|A_{\mathrm{CKA}}'^{-1} X'^\top\| \, \|r' - r\| + \|A_{\mathrm{CKA}}'^{-1}\| \, \|X' - X\| \, \|r\| \\ + \|A_{\mathrm{CKA}}^{-1}\| \, \|A_{\mathrm{CKA}}'^{-1}\| \, \big(\|X'\| + \|X\|\big) \|X' - X\| \, \|X^\top r\|. \end{aligned}$$

Note that both $\|A_{\mathrm{CKA}}^{-1}\|$ and $\|A_{\mathrm{CKA}}'^{-1}\|$ decrease monotonically as $M_{\mathrm{CKA}}$ increases, since they are inverses of positive semidefinite matrices that grow in the Löwner order. As a result, the upper bound on $\|\Delta\theta' - \Delta\theta\|$ monotonically decreases as the CKA weight increases, supporting the intuition in the main text that "distributional sensitivity" is tightened by stronger CKA regularization.

Let $V^\pi$ be the ideal value function, and define the estimation error

$$e(s) = \hat{V}(s) - V^\pi(s).$$

For any state–action pair $(s, a)$, the ideal target and the bootstrapped target are

$$y^\pi(s, a) = r(s, a) + \gamma V^\pi(s'), \qquad y(s, a) = r(s, a) + \gamma \hat{V}(s'),$$

where $s' \sim P(\cdot \mid s, a)$. The bootstrapping error is

$$\varepsilon_{\text{boot}}(s, a) = y(s, a) - y^\pi(s, a) = \gamma\big(\hat{V}(s') - V^\pi(s')\big) = \gamma e(s').$$

Therefore

$$\mathbb{E}\big[\varepsilon_{\text{boot}}^2\big] = \gamma^2 \, \mathbb{E}\big[e(s')^2\big].$$

Let $\hat{V}_{\text{old}}$ and $\hat{V}_{\text{new}}$ denote the value estimates before and after an update, and define

$$e_{\text{old}}(s) = \hat{V}_{\text{old}}(s) - V^\pi(s), \qquad e_{\text{new}}(s) = \hat{V}_{\text{new}}(s) - V^\pi(s).$$

Their difference is the change in the estimated value:

$$e_{\text{new}}(s) - e_{\text{old}}(s) = \hat{V}_{\text{new}}(s) - \hat{V}_{\text{old}}(s) = \Delta V(s).$$

By the linearization and bounds in Secs. D.1.1–D.1.2, one has

$$|\Delta V(s)| \leq \|\omega\| \, \|J_\theta(s)\| \, A(X, \lambda, M_{\text{CKA}}) \, \|r\|,$$

where

$$A(X, \lambda, M_{\text{CKA}}) = \|A_{\text{CKA}}^{-1} X^\top\|.$$

For any state (in particular for $s'$),

$$|e_{\text{new}}(s')| = |e_{\text{old}}(s') + \Delta V(s')| \leq |e_{\text{old}}(s')| + |\Delta V(s')|.$$

Using $(a + b)^2 \leq 2a^2 + 2b^2$, it follows that

$$e_{\text{new}}(s')^2 \leq 2e_{\text{old}}(s')^2 + 2\|\omega\|^2 \|J_\theta(s')\|^2 A(X, \lambda, M_{\text{CKA}})^2 \|r\|^2.$$

Taking expectation over the distribution of $s'$, and defining

$$C_J'^2 := \mathbb{E}\big[\|J_\theta(s')\|^2\big],$$

gives

$$\mathbb{E}\big[e_{\text{new}}(s')^2\big] \leq 2\,\mathbb{E}\big[e_{\text{old}}(s')^2\big] + 2\|\omega\|^2 C_J'^2 A(X, \lambda, M_{\text{CKA}})^2 \|r\|^2.$$

Consequently,

$$\mathbb{E}\big[\varepsilon_{\text{boot,new}}^2\big] = \gamma^2 \, \mathbb{E}\big[e_{\text{new}}(s')^2\big] \leq 2\gamma^2 \, \mathbb{E}\big[e_{\text{old}}(s')^2\big] + 2\gamma^2 \|\omega\|^2 C_J'^2 A(X, \lambda, M_{\text{CKA}})^2 \|r\|^2.$$

Since $A(X, \lambda, M_{\text{CKA}}) = \|A_{\text{CKA}}^{-1} X^\top\|$ decreases monotonically as $M_{\text{CKA}}$ increases (because $A_{\text{CKA}}$ grows in the Löwner order), the second term on the right-hand side tightens as the CKA weight increases. This corresponds to the "second-order upper-bound contraction" on the bootstrapping error stated in Corollary 3.3 of Sec. 3.2.

## D.2. No-Flip Directional Consistency: Expectation-Level Guarantees and One-Step Comparison with PPO

Let the current actor parameters be $\phi$, the policy be $\pi_\phi(\cdot \mid s)$, and the pre-squash mean be $\mu_\phi(s) \in \mathbb{R}^d$. The covariance notation follows Sec. 3. Denote the current diagonal covariance of the policy by

$$\Sigma_{\mathrm{cur}} = \mathrm{diag}(\sigma^2) \succ 0.$$

States in the historical buffer are sampled from a fixed distribution $D_{\mathrm{hist}}$, i.e., $s \sim D_{\mathrm{hist}}$. For each historical sample, define

$$d_{\mathrm{old}}(s) = \mu_{\mathrm{post}}(s) - \mu_{\mathrm{pre}}(s), \qquad d_{\mathrm{new}}(s) = \mu_\phi(s) - \mu_{\mathrm{pre}}(s),$$

where $\mu_{\mathrm{pre}}(s)$ and $\mu_{\mathrm{post}}(s)$ are the pre- and post-update mean actions for that sample under some past "online update," and $\mu_\phi(s)$ is the current mean action. This is consistent with the definitions in Sec. 3.

Under the Mahalanobis geometry induced by $\Sigma_{\mathrm{cur}}$, define the inner product and norm

$$\langle u, v \rangle_{\Sigma_{\mathrm{cur}}^{-1}} = u^\top \Sigma_{\mathrm{cur}}^{-1} v, \qquad \|u\|_{\Sigma_{\mathrm{cur}}^{-1}} = \sqrt{u^\top \Sigma_{\mathrm{cur}}^{-1} u},$$

and the Mahalanobis cosine

$$\cos \theta_M(s) = \frac{\langle d_{\mathrm{new}}(s), d_{\mathrm{old}}(s) \rangle_{\Sigma_{\mathrm{cur}}^{-1}}}{\left( \|d_{\mathrm{new}}(s)\|_{\Sigma_{\mathrm{cur}}^{-1}} \|d_{\mathrm{old}}(s)\|_{\Sigma_{\mathrm{cur}}^{-1}} + \varepsilon \right)},$$

where $\varepsilon > 0$ is a small numerical-stability constant.

Given a threshold angle $\alpha_{\mathrm{thr}} \in [0, \pi/2]$, define $c_{\mathrm{thr}} = \cos(\alpha_{\mathrm{thr}}) \in [0, 1]$. The no-flip hinge loss is

$$L_{\mathrm{noflip}}(\phi) = \mathbb{E}_{s \sim D_{\mathrm{hist}}}\left[ c_{\mathrm{thr}} - \cos \theta_M(s) \right]_+, \qquad [x]_+ = \max(0, x).$$

The geometric proxy metrics are

$$M_1(\phi) = \mathbb{E}\left[\cos \theta_M(s)\right], \qquad M_2(\phi, \delta) = \mathbb{P}\left(\cos \theta_M(s) \leq c_{\mathrm{thr}} - \delta\right), \quad \delta > 0.$$

### D.2.1. LOWER BOUND ON THE EXPECTED COSINE AND UPPER BOUND ON STRONG-VIOLATION PROBABILITY

For any $s$, by the basic property of the hinge function,

$$\left[c_{\mathrm{thr}} - \cos \theta_M(s)\right]_+ \geq c_{\mathrm{thr}} - \cos \theta_M(s).$$

Taking expectations over $s \sim D_{\mathrm{hist}}$ gives

$$L_{\mathrm{noflip}}(\phi) = \mathbb{E}\left[c_{\mathrm{thr}} - \cos \theta_M(s)\right]_+ \geq c_{\mathrm{thr}} - \mathbb{E}\left[\cos \theta_M(s)\right].$$

Rearranging yields

$$M_1(\phi) = \mathbb{E}\left[\cos \theta_M(s)\right] \geq c_{\mathrm{thr}} - L_{\mathrm{noflip}}(\phi),$$

which is exactly the form stated in Lemma 3.4: smaller $L_{\mathrm{noflip}}$ implies a higher lower bound on $M_1$.

For any $\delta > 0$, the hinge property also implies

$$\left[c_{\mathrm{thr}} - \cos \theta_M(s)\right]_+ \geq \delta \cdot \mathbf{1}\{c_{\mathrm{thr}} - \cos \theta_M(s) \geq \delta\},$$

where $\mathbf{1}\{\cdot\}$ is the indicator function. Taking expectation over $s \sim D_{\mathrm{hist}}$ yields

$$L_{\mathrm{noflip}}(\phi) \geq \delta \cdot \mathbb{P}\left(c_{\mathrm{thr}} - \cos \theta_M(s) \geq \delta\right) = \delta \cdot \mathbb{P}\left(\cos \theta_M(s) \leq c_{\mathrm{thr}} - \delta\right).$$

Dividing both sides by $\delta > 0$ gives

$$M_2(\phi, \delta) = \mathbb{P}\left(\cos \theta_M(s) \leq c_{\mathrm{thr}} - \delta\right) \leq \frac{L_{\mathrm{noflip}}(\phi)}{\delta},$$

which corresponds to Lemma 3.4: smaller $L_{\mathrm{noflip}}$ yields a smaller upper bound on the strong-violation probability (i.e., cases where the angle is significantly larger than $\alpha_{\mathrm{thr}}$), and the bound decays in a $1/\delta$ fashion for the tail.

D.2.2. ONE-STEP COMPARISON WITH PPO

Let $L_{\text{PPO}}(\phi)$ denote the standard clipped PPO actor loss, and define its update direction

$$v_{\text{PPO}}(\phi) = -\nabla_\phi L_{\text{PPO}}(\phi).$$

With the no-flip term added, the total loss becomes

$$L_{\text{total}}(\phi) = L_{\text{PPO}}(\phi) + \lambda_{\text{noflip}} L_{\text{noflip}}(\phi), \qquad \lambda_{\text{noflip}} > 0,$$

with update direction

$$v_{\text{total}}(\phi) = -\nabla_\phi L_{\text{total}}(\phi) = -\nabla_\phi L_{\text{PPO}}(\phi) - \lambda_{\text{noflip}} \nabla_\phi L_{\text{noflip}}(\phi).$$

Let $\alpha > 0$ be the learning rate. A first-order Taylor expansion of $L_{\text{noflip}}$ at $\phi$ gives:

**One step of PPO:**

$$\begin{aligned} L_{\text{noflip}}(\phi + \alpha v_{\text{PPO}}) &= L_{\text{noflip}}(\phi) + \alpha \left\langle \nabla_\phi L_{\text{noflip}}(\phi), v_{\text{PPO}}(\phi) \right\rangle + o(\alpha) \\ &= L_{\text{noflip}}(\phi) - \alpha \left\langle \nabla_\phi L_{\text{noflip}}(\phi), \nabla_\phi L_{\text{PPO}}(\phi) \right\rangle + o(\alpha). \end{aligned}$$

**One step with the no-flip term:**

$$\begin{aligned} L_{\text{noflip}}(\phi + \alpha v_{\text{total}}) &= L_{\text{noflip}}(\phi) + \alpha \left\langle \nabla_\phi L_{\text{noflip}}(\phi), v_{\text{total}}(\phi) \right\rangle + o(\alpha) \\ &= L_{\text{noflip}}(\phi) - \alpha \left\langle \nabla_\phi L_{\text{noflip}}(\phi), \nabla_\phi L_{\text{PPO}}(\phi) \right\rangle - \alpha \lambda_{\text{noflip}} \| \nabla_\phi L_{\text{noflip}}(\phi) \|^2 + o(\alpha), \end{aligned}$$

where $\langle g, g \rangle = \|g\|^2$ is used in the last term.

Subtracting the two expansions yields

$$L_{\text{noflip}}(\phi + \alpha v_{\text{total}}) = L_{\text{noflip}}(\phi + \alpha v_{\text{PPO}}) - \alpha \lambda_{\text{noflip}} \| \nabla_\phi L_{\text{noflip}}(\phi) \|^2 + o(\alpha).$$

Applying the lower bound from Sec. D.2.1,

$$M_1(\phi) \geq c_{\text{thr}} - L_{\text{noflip}}(\phi),$$

to the points $\phi + \alpha v_{\text{PPO}}$ and $\phi + \alpha v_{\text{total}}$ gives

$$\begin{aligned} M_1(\phi + \alpha v_{\text{total}}) &\geq c_{\text{thr}} - L_{\text{noflip}}(\phi + \alpha v_{\text{total}}) \\ &= c_{\text{thr}} - L_{\text{noflip}}(\phi + \alpha v_{\text{PPO}}) + \alpha \lambda_{\text{noflip}} \| \nabla_\phi L_{\text{noflip}}(\phi) \|^2 - o(\alpha), \end{aligned}$$

while

$$M_1(\phi + \alpha v_{\text{PPO}}) \geq c_{\text{thr}} - L_{\text{noflip}}(\phi + \alpha v_{\text{PPO}}).$$

Combining the two gives the first-order comparison

$$M_1(\phi + \alpha v_{\text{total}}) \gtrsim M_1(\phi + \alpha v_{\text{PPO}}) + \alpha \lambda_{\text{noflip}} \| \nabla_\phi L_{\text{noflip}}(\phi) \|^2,$$

which corresponds to the "expected-alignment improvement" part of Proposition 3.5: starting from the same point and using the same step size, adding the no-flip term yields an additional increase in the lower bound of the directional-alignment metric.

Similarly, for the strong-violation probability, the upper bound from Sec. D.2.1,

$$M_2(\phi, \delta) \leq \frac{L_{\text{noflip}}(\phi)}{\delta},$$

applied at $\phi + \alpha v_{\text{PPO}}$ and $\phi + \alpha v_{\text{total}}$, gives

$$\begin{aligned} M_2(\phi + \alpha v_{\text{total}}, \delta) &\leq \frac{L_{\text{noflip}}(\phi + \alpha v_{\text{total}})}{\delta} \\ &= \frac{L_{\text{noflip}}(\phi + \alpha v_{\text{PPO}}) - \alpha \lambda_{\text{noflip}} \| \nabla_\phi L_{\text{noflip}}(\phi) \|^2 + o(\alpha)}{\delta}, \end{aligned}$$

whereas

$$M_2(\phi + \alpha v_{\text{PPO}}, \delta) \leq \frac{L_{\text{noflip}}(\phi + \alpha v_{\text{PPO}})}{\delta}.$$

Comparing the two yields, at the first-order level,

$$M_2(\phi + \alpha v_{\text{total}}, \delta) \lesssim M_2(\phi + \alpha v_{\text{PPO}}, \delta) - \frac{\alpha \lambda_{\text{noflip}}}{\delta} \|\nabla_\phi L_{\text{noflip}}(\phi)\|^2,$$

which corresponds to the "tightening of the strong-violation probability upper bound" stated in Proposition 3.5.

### D.3. KDE-Based Novelty Shaping: Improvement of Occupancy Mass

Unless otherwise stated, notation follows Sec. 3. To avoid confusion with the bandwidth symbol $b > 0$ used in Sec. 3.4, the kernel bandwidth is denoted by $\sigma > 0$ in this section, while the baseline constant continues to use the symbol $b$.

#### D.3.1. LOCAL GRADIENT LOWER BOUND OF THE KDE NOVELTY FIELD

Within a single PPO update, let the set of online states be $\{s_t\}_{t=1}^n$, and denote the successor state by $s' = f(s, a)$ with latent representation

$$h' := h_\theta(s') \in \mathbb{R}^H.$$

Consider the Gaussian kernel

$$k_\sigma(u, v) = \exp\left(-\frac{\|u - v\|^2}{2\sigma^2}\right),$$

and construct a kernel density estimator over a set of reference points $\{z_j\}_{j=1}^m$:

$$\hat{p}(h') = \frac{1}{m} \sum_{j=1}^m k_\sigma(h', z_j),$$

with the latent-space novelty defined as

$$N(h') = -\log \hat{p}(h').$$

Taking the gradient with respect to $h'$ and using

$$\nabla_{h'} k_\sigma(h', z_j) = -\frac{1}{\sigma^2} k_\sigma(h', z_j)(h' - z_j),$$

one obtains

$$\nabla_{h'} N(h') = -\frac{1}{\hat{p}(h')} \nabla_{h'} \hat{p}(h') = \frac{1}{\sigma^2} \frac{\sum_{j=1}^m k_\sigma(h', z_j)(h' - z_j)}{\sum_{j=1}^m k_\sigma(h', z_j)}.$$

Define

$$\omega_j = \frac{k_\sigma(h', z_j)}{\sum_{l=1}^m k_\sigma(h', z_l)}, \qquad v_j = \frac{h' - z_j}{\|h' - z_j\|},$$

so that

$$\nabla_{h'} N(h') = \frac{1}{\sigma^2} \sum_{j=1}^m \omega_j \|h' - z_j\| v_j.$$

Suppose there exist a constant $\rho > 0$ and an opening angle $\alpha \in (0, \pi/2]$, and a region $\Omega_\varepsilon \subset \mathbb{R}^H$, such that for any $h \in \Omega_\varepsilon$:

1. All reference points are at least distance $\rho$ away:

$$\min_j \|h - z_j\| \geq \rho;$$

2. There exists a unit vector $u$ such that every direction $v_j$ lies within an angular cone of half-angle $\alpha$ around $u$, i.e.,

$$\langle v_j, u \rangle \geq \cos \alpha > 0.$$

Then for any $h \in \Omega_\varepsilon$, the directional derivative of $N$ in direction $u$ satisfies

$$\langle \nabla_{h'} N(h), u \rangle = \frac{1}{\sigma^2} \sum_{j=1}^m \omega_j \, \|h - z_j\| \, \langle v_j, u \rangle \geq \frac{1}{\sigma^2} \sum_{j=1}^m \omega_j \, \rho \, \cos \alpha = \frac{\rho \cos \alpha}{\sigma^2}.$$

Hence

$$\|\nabla_{h'} N(h)\| \geq \langle \nabla_{h'} N(h), u \rangle \geq c_*, \qquad c_* := \frac{\rho \cos \alpha}{\sigma^2} > 0.$$

This shows that, within a region where the reference points "surround" $h$ in a cone-like configuration, the novelty field $N$ has a uniformly positive lower bound on its directional derivative along $u$, and thus does not degenerate to zero gradient in low-density regions.

### D.3.2. Exponential Tilting of the Occupancy Measure and Improvement in Low-Density Regions

Let $\pi_\phi$ be a policy and let $\rho_\phi$ denote its induced state–action occupancy measure. For notational simplicity write $\rho \equiv \rho_\phi$, and use

$$\langle \rho, f \rangle = \int f(x) \, \rho(dx)$$

for the integral of a function $f$ with respect to $\rho$. Define a novelty field in latent space by

$$g(x) := N\big(h_\theta(f(s, a))\big), \qquad x = (s, a),$$

so that the novelty-related objective can be written as

$$J_{\mathrm{nov}}(\phi) = \beta \langle \rho_\phi, g \rangle,$$

where $\beta > 0$ corresponds (to first order) to the novelty-shaping strength in the main text (analogous to $\lambda_{\mathrm{nov}}$).

Consider the following variational update in occupancy space: given the current occupancy measure $\rho$, define the next occupancy $\rho^+$ as the solution to

$$\max_{\rho^+ \in \mathcal{P}(X)} \left\{ \beta \langle \rho^+, g \rangle - \frac{1}{\eta} \mathrm{KL}(\rho^+ \, \| \, \rho) - \beta b \langle \rho^+, 1 \rangle \right\}, \quad \text{s.t. } \langle \rho^+, 1 \rangle = 1,$$

where $\mathcal{P}(X)$ is the set of probability measures on $X$, $\eta > 0$ is a step-size/temperature parameter, $b$ is a baseline, and $\mathrm{KL}(\rho^+ \, \| \, \rho)$ is the Kullback–Leibler divergence. The first term encourages novelty through $g$, the second term penalizes deviations from $\rho$, and the third term is a linear baseline term (which does not change the structure of the optimizer but affects the normalization constant).

The Lagrangian of this constrained problem is

$$L(\rho^+, \lambda) = \beta \int g(x) \, \rho^+(dx) - \frac{1}{\eta} \int \rho^+(x) \log \frac{\rho^+(x)}{\rho(x)} \, dx - \beta b \int \rho^+(dx) - \lambda \left( \int \rho^+(dx) - 1 \right).$$

Taking the functional derivative with respect to $\rho^+$ and setting it to zero (KKT condition) gives

$$\frac{\delta L}{\delta \rho^+}(x) = \beta g(x) - \frac{1}{\eta} \left( \log \frac{\rho^+(x)}{\rho(x)} + 1 \right) - \beta b - \lambda = 0.$$

Rearranging and exponentiating yields

$$\log \frac{\rho^+(x)}{\rho(x)} = \eta \beta \big( g(x) - b \big) - \eta \lambda - 1 \quad \Longrightarrow \quad \frac{\rho^+(x)}{\rho(x)} = C \exp \big( \eta \beta \big( g(x) - b \big) \big),$$

where $C = e^{-1 - \eta \lambda}$ does not depend on $x$. Imposing the normalization constraint $\int \rho^+ = 1$ yields

$$C = \frac{1}{Z}, \qquad Z = \int \exp \big( \eta \beta (g(x) - b) \big) \, \rho(dx) = \langle \rho, \exp(\eta \beta (g - b)) \rangle.$$

Thus the updated occupancy has the closed form

$$\rho^+(dx) = \frac{\exp \big( \eta \beta (g(x) - b) \big)}{Z} \, \rho(dx), \qquad Z = \langle \rho, \exp(\eta \beta (g - b)) \rangle,$$

i.e., the occupancy distribution is obtained by an exponential tilting of $\rho$ along the novelty field $g$.

### D.3.3. Non-Decreasing Occupancy Mass in High-Novelty (Low-Density) Regions

Let $\tau_0 \in \mathbb{R}$ be a threshold, and define the "high-novelty" set

$$A := \{x \in X : g(x) \geq \tau_0\}, \qquad A^c = X \setminus A.$$

Let the indicator function be

$$\mathbf{1}_A(x) = \begin{cases} 1, & x \in A, \\ 0, & x \notin A, \end{cases}$$

and denote the original occupancy mass of $A$ by

$$p := \langle \rho, \mathbf{1}_A \rangle = \mathbb{P}_\rho(A) \in [0,1],$$

and the updated mass by

$$m^+ := \langle \rho^+, \mathbf{1}_A \rangle.$$

Define the weight function

$$\Phi(x) := \exp\big(\eta\beta(g(x) - b)\big),$$

then by the exponential-tilting form,

$$m^+ = \int \mathbf{1}_A(x)\, \rho^+(dx) = \frac{\langle \rho, \Phi \mathbf{1}_A \rangle}{\langle \rho, \Phi \rangle} = \frac{\mathbb{E}_\rho[\Phi\, \mathbf{1}_A]}{\mathbb{E}_\rho[\Phi]}.$$

Let

$$u := \mathbb{E}_\rho[\Phi \mid A], \qquad v := \mathbb{E}_\rho[\Phi \mid A^c],$$

then by the law of total expectation,

$$\mathbb{E}_\rho[\Phi\, \mathbf{1}_A] = up, \qquad \mathbb{E}_\rho[\Phi] = up + v(1-p),$$

and hence

$$m^+ = \frac{up}{up + v(1-p)}.$$

Assume the baseline satisfies $b \leq \tau_0$. Then for $x \in A$,

$$g(x) - b \geq \tau_0 - b \geq 0 \quad \Longrightarrow \quad \Phi(x) \geq \exp\big(\eta\beta(\tau_0 - b)\big),$$

while for $x \in A^c$,

$$g(x) - b < \tau_0 - b \quad \Longrightarrow \quad \Phi(x) \leq \exp\big(\eta\beta(\tau_0 - b)\big).$$

Thus

$$\min_{x \in A} \Phi(x) \geq \max_{x \in A^c} \Phi(x),$$

which implies

$$u = \mathbb{E}_\rho[\Phi \mid A] \geq \min_{x \in A} \Phi(x) \geq \max_{x \in A^c} \Phi(x) \geq \mathbb{E}_\rho[\Phi \mid A^c] = v,$$

and hence $u \geq v$.

Comparing $m^+$ and $p$, one has

$$m^+ - p = \frac{up}{up + v(1-p)} - p = \frac{p(1-p)(u-v)}{up + v(1-p)}.$$

The denominator $up + v(1-p) > 0$, while $p(1-p) \geq 0$ and $u - v \geq 0$, so

$$m^+ - p \geq 0 \quad \Longrightarrow \quad m^+ \geq p = \mathbb{P}_\rho(A).$$

When $p \in (0,1)$ and $\rho(\{x : g(x) = \tau_0\}) < 1$, both $A$ and $A^c$ have positive measure and the exponential weights differ across them, implying $u > v$ and thus $m^+ > p$.

Therefore, as long as the baseline satisfies $b \leq \tau_0$, the exponential tilting guarantees that the occupancy mass of the high-novelty set $A$ does not decrease after one update, and is strictly increased in nontrivial cases. This matches Theorem 3.6 in Sec. 3.4, and shows that adding truncated novelty shaping into the advantage is equivalent to placing extra weight on low-density regions in the "value-related" latent space, thereby increasing their visitation probability from the perspective of the occupancy measure.

