# OpenReview forum: "Stabilizing PPO via Latent-Space Regularization and KDE-Driven Exploration"
_ICML.cc/2026/Conference — ICML 2026 regular_

### Official Review · Reviewer_5bGu · 2026-03-10

**Soundness:** 4
**Presentation:** 4
**Significance:** 3
**Originality:** 3
**Overall Recommendation:** 5
**Confidence:** 4

**Summary:**

This paper proposes three techniques from different sides to enhance PPO's performance and conducts comprehensive experiments to evaluate them.

**Compliance With Llm Reviewing Policy:**

Affirmed.

**Final Justification:**

I maintain my Accept recommendation after considering the rebuttal. The paper is technically sound, clearly written, and supported by strong experiments and ablations. Its main contribution is a practical improvement to PPO with useful empirical insights, though the originality is moderate. The limitations are the added hyperparameter burden and the interaction among the three components.

**Key Questions For Authors:**

1. Is it possible to do some statistical analysis or causal identification of the interaction between each component? I find that some components might have contradictory effects. For example, CKA helps stability but might be conservative and limit exploration, while KDE could encourage exploration. There might be some trade-offs among these techniques.
2. In Appendix B.6, it shows that SPPO takes about 2 hours but PPO takes about 3 hours, and it also mentions that the wall-clock difference is mainly due to the KDE computation. So why would KDE reduce the time?
3. In Sec. 4, it says the baselines share identical architectures, optimizers, and general HPs. But I think the baselines might need their own tuned hyperparameters to show their actual performance.


Typos:
1.  “Combining this relation with Lemma 5.4 yields” — is it Lemma 3.4?
2. Does Table 8 list parameter combinations used in Appendix C.6 or Appendix C.3? The caption of Table 8 says C.3, but Appendix B.4 says C.6.

**Limitations:**

Yes

**Strengths And Weaknesses:**

**Strengths**
1. I like the experiments. They are comprehensive and evaluate the effect of each technique, offering some insights into which factors are important for performance.

**Weaknesses**
1. The method introduces 9 HPs, which is nontrivial, although the paper says the default setting lies near a stable region for the tested tasks. From the experiments, some tasks are sensitive to the HPs. Also, although it provides sensitivity analyses, it analyzes each one while keeping the others fixed, but the HPs might jointly affect performance. Anyway, it is still a heavy burden.
2. I think the three techniques are each good, and the ablation studies support them. But it is not clear whether the components have interactive effects or not.

---

> ### Author Rebuttal · Authors · 2026-03-29
>
> Thank you for the careful review. The key questions and the weaknesses are addressed below.
>
> Q1
>
> Thank you for this question. The three components do not combine in a purely linear manner, but exhibit clear interaction effects. Section 4.2 compares full SPPO, variants with one component removed, and variants with only one component retained, and the observed gains are consistent with the coupling among representation stability, update consistency, and exploration behavior. Potential trade-offs are also acknowledged. In particular, stronger CKA regularization may improve stability by suppressing critic representation drift, but may also hinder adaptation when the policy distribution changes rapidly, while KDE mainly helps alleviate under-exploration. Appendix C.6 likewise suggests that different environments prefer different regularization strengths. For a more detailed discussion of these interaction effects and the role of KDE in the full combination, please also see the responses to Reviewer cyHV, Q1, and Reviewer J2H8, Q1. In the final version, this discussion will be further strengthened, and the contribution of each component will be clarified more explicitly.
>
> Q2
>
> Thank you for pointing out this ambiguity. There is a typo in Appendix B.6. For $\texttt{Humanoid-v5}$, a single PPO seed takes about 2 hours, while a single SPPO seed takes about 3 hours, mainly due to the additional KDE computation. Please also see the response to Reviewer cyHV, Q3, for a more complete discussion of the runtime overhead. This typo will be corrected, and the training-time differences will be clarified more explicitly in the final version.
>
> Q3
>
> Thank you for raising this point. In the current setup, PPO, SPPO, CHAIN, PEER, and PFO share the same network architecture, optimizer, and general training hyperparameters, so that the comparison focuses on algorithmic mechanisms rather than large-scale tuning advantages. The original PPO setup was first tuned and then used as the common starting point for PPO, the three baselines, and SPPO. The method-specific hyperparameters of the other three baselines were set according to their original papers and implementations, and kept the same across environments. SPPO was evaluated under the same comparison setting. The purpose of this design is not to favor any specific algorithm, but to evaluate all methods under a fair and controlled setting.
>
> Further tuning of the three baselines for each environment could potentially yield higher performance than that shown in the current manuscript, but the same is also true for SPPO, since the default SPPO hyperparameters in Appendix C.6 are often not optimal for every environment. The present comparison is therefore intended as a controlled comparison rather than an exhaustive tuning comparison. In the final version, this experimental goal and boundary will be clarified more explicitly.
>
> Weakness 1: the burden of 9 additional hyperparameters
>
> Thank you for pointing out this issue. Appendix B.4 explicitly lists the 9 additional hyperparameters introduced by SPPO relative to PPO, and Appendix C.6 provides sensitivity analysis around the default setting. The results show that the default setting is generally located in a relatively stable region for most environments, although preferences do differ across environments; for example, $\texttt{Hopper-v5}$ is more sensitive, while $\texttt{Walker2d-v5}$ sometimes performs slightly better under weaker regularization.
> At the same time, these 9 hyperparameters do not have equal tuning priority. The three weight parameters, $\lambda_\mathrm{cka}$, $\lambda_\mathrm{nf}$, and $\lambda_\mathrm{kde}$, are the main control variables and should be tuned first, while the remaining hyperparameters can generally be kept at their default values. The main purpose of Appendix C.6 is therefore not exhaustive hyperparameter search, but to show how each parameter affects training and final return, thereby clarifying the sensitivity of SPPO. It is agreed that joint sensitivity under multi-parameter tuning deserves deeper analysis. However, the current sensitivity analysis is mainly intended to demonstrate the usability of the default configuration and local robustness, rather than to fully characterize interactions in a high-dimensional hyperparameter space. This point will be clarified more explicitly in the final version.
>
> Weakness 2: whether the three techniques have interaction effects
>
> Thank you for emphasizing this point. This concern is essentially the same as Q1 above, where the nonlinear interaction among the three components has already been clarified. The full discussion is therefore not repeated here.
>
> On the two typo/cross-reference issues
>
> Thank you for pointing out these two issues. The lemma reference should be Lemma 3.4 rather than Lemma 5.4, and the cross-reference for Table 8 should be Appendix C.6 rather than Appendix C.3. Both will be corrected in the final version.

---

> > ### Author Rebuttal · Reviewer_5bGu · 2026-04-02
> >
> > Thank the authors for the rebuttal, and I will keep my score as 5.

---

> > > ### Author Response · Authors · 2026-04-03
> > >
> > > Thank you for the encouraging response and for taking the time to review the rebuttal. The maintained score is greatly appreciated.

---

### Official Review · Reviewer_J2H8 · 2026-03-11

**Soundness:** 3
**Presentation:** 2
**Significance:** 3
**Originality:** 2
**Overall Recommendation:** 4
**Confidence:** 4

**Summary:**

This paper introduces Stabilized PPO (SPPO), a set of modifications to the Proximal Policy Optimization (PPO) algorithm for continuous control tasks. The authors first observe through CKA (Centered Kernel Alignment) analysis that the latent representations of both actor and critic networks undergo substantial structural changes and direction flips during training, even after the policy appears to stabilize. To address these issues, SPPO adds three components to PPO: (i) a CKA-based regularizer that encourages consistency of critic latent representations over short sequences, (ii) a “no-flip” penalty that penalizes large directional changes in the actor’s pre-squash mean action relative to past updates, and (iii) a KDE-based novelty shaping term that adjusts advantages to encourage exploration in low-density regions of the critic latent space. The authors provide theoretical analysis showing that the CKA regularizer tightens upper bounds on representation drift and bootstrapping error. Experiments on MuJoCo, DeepMind Control Suite, and Atari tasks demonstrate improved sample efficiency and final performance compared to PPO and several recent variants (CHAIN, PEER, PFO). Ablation studies and churn metrics are used to analyze the contribution of each component.

**Compliance With Llm Reviewing Policy:**

Affirmed.

**Final Justification:**

My concerns are solved, and hence I would like to raise my score to 4.

**Key Questions For Authors:**

1. What is the logical necessity of the KDE exploration component? It appears orthogonal to the two stability issues (critic representation drift and actor direction flips) that motivate the paper.

2. The theoretical derivations in Appendix D rely on several key assumptions (linearization, small update step, bounded covariance of residuals, etc.). Could you explain in intuitive terms: To what extent do these assumptions hold in practical deep RL training? If they are not strictly satisfied, how much of the theoretical guarantee can still be expected to hold?

3. In Figure 1, are the curves shown averaged over multiple runs or from a single run?

4. Some writing issues:
- Please provide the full names of CKA and KDE in the abstract to help readers from adjacent fields.
- Figure 2 is difficult to read: what do the points, lines, and shaded regions in each subfigure represent? Please list all elements explicitly in the caption.

**Limitations:**

Yes

**Strengths And Weaknesses:**

**Strengths**

- Empirical grounding: The work is motivated by a clear empirical observation (CKA trajectories in Fig. 1) that reveals instability in intermediate representations—a valuable insight in itself.

- Extensive experimentation: The evaluation covers multiple continuous-control benchmarks (MuJoCo, DMC) and even extends to discrete-action Atari games. Comparisons with several recent PPO variants (CHAIN, PEER, PFO) are included, along with thorough ablation studies and hyperparameter sensitivity analysis (Appendix C). The use of Rliable metrics for aggregate statistics is a plus.

- Theoretical effort: The authors attempt to provide theoretical justification for the CKA regularizer, showing that it leads to tighter bounds on pointwise representation drift and one-step bootstrapping error (Propositions 3.1–3.3, Corollary 3.3). This adds a layer of rigor beyond purely empirical contributions.

- Reproducibility details: Hyperparameters, network architectures, and implementation details are provided in the appendix, which should facilitate reproduction.

**Weakness**

- KDE-driven novelty shaping is added without a clear connection to the observed instability problems. While the authors mention it is meant to counteract under-exploration that might arise from the other regularizers, this rationale feels post hoc and makes the overall contribution appear like a collection of loosely related heuristics rather than a unified solution.

- Unclear problem definition: Despite the empirical motivation, the paper never crisply defines what “instability” or “flipping” means in a way that a practitioner can immediately grasp. The writing is dense and relies heavily on undifined terminologies (e.g., “latent space geometry,” “representational drift,” “subspace changes”), making it difficult for readers outside a narrow subfield to appreciate the core issue.

- Theoretical analysis rests on strong assumptions: The derivations in Appendix D rely on linearization, small updates, and frozen parameters, which are not guaranteed to hold in practice. While such analysis can be insightful, its practical relevance is unclear, and it does not convincingly explain why the combined method works as well as it does.

---

> ### Author Rebuttal · Authors · 2026-03-29
>
> Thank you for the careful review and constructive comments. The key questions and general concerns are answered below.
>
> Q1
>
> Thank you for raising this point. KDE is not a third stability constraint, but an exploration-enhancing component whose standalone gain is relatively mild and mainly appears in the early and middle stages of training. Its main value is to compensate for the exploration conservatism induced by stabilization. When combined with only one of the other two components, its gain remains limited. Its benefit becomes more evident only in full SPPO, where actor-side stabilization, critic-side stabilization, and exploration shaping work together. A more complete discussion is provided in the response to Reviewer cyHV, Q1. In the final version, the functional role of KDE will be clarified more explicitly.
>
> Q2
>
> Thank you for this question. Appendix D does rely on several assumptions, including a small update step, first-order local linearization, frozen external normalization within the current update, and, in part of the derivation, bounded residual covariance. However, these derivations are not intended as a strict global description of full deep nonlinear training dynamics, but as a local mechanistic analysis of how the proposed regularizers can improve conditioning and tighten upper bounds on representation drift and bootstrapping error under local approximation. This type of treatment is a common theoretical tool in deep network optimization, and remains reasonable here because the critic update can still be viewed essentially as a regression process around TD targets.
>
> Accordingly, these assumptions are regarded as analytically meaningful for local single-step explanation. The role of the theory is to explain why the proposed regularizers should reduce representation drift, improve update conditioning, and suppress bootstrap error accumulation under such analyzable conditions. The more important question is whether these mechanisms have observable empirical counterparts. This is why Section 4.3 constructs across-update metrics and shows across multiple environments that SPPO yields lower critic CKA churn, narrower policy variability, and smoother actor-critic update behavior than PPO. These observations are consistent with the mechanisms captured in Appendix D.
>
> More precisely, Appendix D provides theoretical support for local mechanisms and trends, while the effective operating regime and optimal regularization strength in real deep RL training still need to be characterized empirically. This theoretical positioning will be made clearer in the final version.
>
> Q3.
>
> Thank you for this question. Figure 1 is based on a single run for each of the five environments and is intended as a within-run training-dynamics diagnostic rather than a seed-aggregated performance summary. To address seed variability, this diagnostic was additionally repeated over six random seeds, and the mean trajectory with the min-max envelope was plotted and will be updated in the appendix of the final version (For visual reference only, an anonymous link is provided here: https://edytbetrit.github.io/figure/). The aggregated results show the same qualitative pattern as in the original figure, namely that critic-side late-layer CKA generally stabilizes later than actor-side CKA across the considered environments, suggesting that the observation is not specific to a selected run.
>
> Q4.
>
> Thank you for pointing out these issues. The full names of CKA and KDE should indeed be given at their first appearance in the abstract, and this will be corrected in the final version.
>
> The comment on the readability of Figure 2 is also appreciated. The points, lines, and shaded regions are mainly used to illustrate the roles of the three proposed components. In the final version, a fuller caption will be provided, and the meaning of each visual element in all three subfigures will be explained explicitly.
>
> On the unclear problem definition.
>
> Thank you for pointing out this concern. In the final version, instability will be defined more explicitly as the oscillatory value, policy, and representation changes of the actor and critic across consecutive updates, and flipping as large directional reversals of actor pre-squash action updates relative to the historical improvement direction.
>
> On whether the method is a unified solution.
>
> This concern is well taken. Although the three components differ in form, they are designed around a unified objective: reducing update oscillation and actor-critic mismatch while preserving effective exploration under improved stability. Under this view, they correspond to three complementary aspects of actor-critic training dynamics rather than unrelated heuristics for separate issues. This is also reflected in the method design in Section 3 and the ablations in Section 4.2. This unified complementary-mechanism framing will be strengthened in the final version.

---

> > ### Author Rebuttal · Reviewer_J2H8 · 2026-04-02
> >
> > Thank you for your rebuttal, which partially addresses my concern. However, I still have reservations about the problem definition, which the authors have promised to further clarify in the revision. Could the authors provide a possible revision at this rebuttal stage?

---

> > > ### Author Response · Authors · 2026-04-04
> > >
> > > Thank you for the follow-up question. A more explicit explanation of the problem definition considered in this paper is provided below, together with a revision paragraph to be added in the final revised manuscript.
> > >
> > > In this paper, $\texttt{instability}$ refers to a form of oscillation in the actor-critic update process. More specifically, it refers to the presence of unnecessary or non-smooth changes in the critic’s value predictions, the critic’s latent representations, or the actor’s pre-squash mean actions, which can be examined by comparing the actor and critic before and after two consecutive updates on a fixed set of states (used as probes to diagnose the change in the network). Here, $\texttt{latent space geometry}$ refers to the relative arrangement of probe-state representations in the latent space; $\texttt{representational drift}$ denotes the change of latent features for the same probe states across updates; and $\texttt{subspace changes}$ denote changes in the principal span or similarity structure of those representations, which are quantified by CKA.
> > >
> > > Beyond presenting a representation analysis result, Fig. 1 also serves to illustrate this specific phenomenon. That is, although the agent may already have entered a relatively stable training stage, substantial restructuring may still persist across consecutive updates, especially in the critic’s latent representations and the actor’s pre-squash outputs. This paper argues that such non-smooth changes may increase actor-critic mismatch, weaken the consistency between value estimation and policy improvement, and further affect training efficiency and stability. The diagnostic metrics constructed in Section 4.3 are, in fact, used as tools to evaluate the stability of the network update, including value change, value churn, CKA, CKA churn, and policy-output change.
> > >
> > > The term $\texttt{flipping}$ used in this paper refers to a large directional reversal of the actor’s current pre-squash mean-action change relative to the historical improvement direction on the same state. In other words, it describes a situation in which a historical update pushes the pre-squash mean corresponding to a given state in one direction, but a later update pushes it in the opposite direction. The no-flip regularizer is thus introduced to restrain such strongly direction-reversing updates, and this definition is also consistent with the formal construction of “No-Flip Penalty” in Section 3.3 of the manuscript.
> > >
> > > In the revised manuscript, a short explanatory paragraph on the problem definition is added in the introduction, after the discussion of the phenomenon shown in Fig. 1 and before the formal introduction of SPPO. The intended logic is first to present the empirical observation, then to make explicit what is regarded in this paper as the problem to be addressed, and then to transition naturally to the introduction of the method.
> > >
> > > The revision paragraph is as follows:
> > >
> > > “In this paper, instability refers specifically to oscillation of the network update during training rather than performance fluctuation in a broad sense. Given a fixed set of probe states, instability occurs when two consecutive updates introduce unnecessary or non-smooth changes in the critic’s value predictions, the critic’s latent representations, or the actor’s pre-squash mean actions. Representational drift describes how the critic’s latent features change across updates on the same probe states. Flipping refers to a large directional reversal of the actor’s current pre-squash mean-action change relative to the improvement direction on the same state. Such oscillation may increase actor-critic mismatch and weaken the consistency between value estimation and policy improvement. The goal of SPPO is therefore to reduce these unnecessary across-update oscillations while still preserving sufficient exploration.”
> > >
> > > The above revision is intended to make the paper clearer in three aspects. First, it makes explicit that the problem targeted in this paper is across-update oscillation rather than broad return variance. Second, it clearly distinguishes instability as the broader problem from flipping as the more specific actor-side phenomenon. Third, it makes it easier for readers to see that the diagnostic metrics in Section 4.3 are not post hoc analyses, but empirical characterizations that directly correspond to the problem definition introduced earlier in the paper.
> > >
> > > Thank you again for raising this point. It is believed that the above clarification will help improve the readability of the paper and the clarity of its overall argument.

---

### Official Review · Reviewer_cyHV · 2026-03-13

**Soundness:** 3
**Presentation:** 3
**Significance:** 3
**Originality:** 3
**Overall Recommendation:** 5
**Confidence:** 3

**Summary:**

The paper proposes SPPO, a stabilised variant of Proximal Policy Optimisation designed to improve training stability and sample efficiency in reinforcement learning. The method introduces three modifications to the standard PPO. First, the authors propose CKA-based critic latent regularisation, which constrains changes in the critic’s internal representations across updates using Centered Kernel Alignment, aiming to stabilise value estimation. Second, they introduce a no-flip penalty for actor updates that penalises policy updates moving in the opposite direction of the advantage signal, encouraging policy changes that are consistent with the estimated advantage. Third, the method incorporates KDE-based novelty shaping to promote exploration by providing intrinsic rewards for visiting rarely observed states.

The authors provide theoretical motivation for the critic regularisation by analysing the critic update as a ridge regression problem and showing that the CKA regulariser adds a positive semidefinite term to the normal matrix, which improves conditioning. Empirically, the proposed approach is evaluated on several continuous control benchmarks, including MuJoCo environments, and compared with PPO and several stabilisation baselines. The results suggest that the proposed modifications can improve training stability and achieve higher final performance across multiple tasks.

**Compliance With Llm Reviewing Policy:**

Affirmed.

**Key Questions For Authors:**

1- The proposed method combines three modifications to PPO: CKA-based critic latent regularisation, the no-flip penalty for actor updates, and KDE-based novelty shaping for exploration. Could the authors provide more detailed ablation results isolating the individual contribution of each component across the reported benchmarks? In particular, it would be useful to understand whether the observed performance improvements are primarily driven by one component or by the interaction of all three. Clarifying this would help assess the necessity and relative importance of each component.
2- The theoretical motivation for the CKA-based critic regularisation relies on a ridge-regression interpretation of the critic update and assumes a linearised feature representation. Could the authors clarify how well this approximation reflects the behaviour of the deep critic networks used in the experiments? Additional discussion or empirical evidence showing that the representation stability enforced by CKA correlates with improved value estimation or advantage quality would strengthen the theoretical justification.
3- The KDE-based novelty shaping introduces a density estimation step over the visited states. How does the computational cost of this component scale with the dimensionality of the state space and the number of collected samples? It would be helpful if the authors could report the additional computational overhead introduced by this component and discuss whether the approach remains practical for higher-dimensional environments.

**Limitations:**

The paper could benefit from a more explicit discussion of the practical limitations of the proposed components. For example, the KDE-based novelty shaping may introduce additional computational overhead and may not scale well to high-dimensional state spaces, which could limit its applicability in more complex environments. Similarly, the CKA-based critic regularisation assumes that constraining representation changes improves stability, but this may also restrict the critic’s ability to learn new representations when the policy distribution shifts significantly. The no-flip penalty may also introduce bias if the advantage estimates are noisy or inaccurate, potentially preventing beneficial policy updates.

**Strengths And Weaknesses:**

The paper addresses the important and well-recognised challenge of improving the training stability of PPO. The use of representation similarity through CKA for critic regularisation is an interesting idea that connects reinforcement learning with techniques from representation learning and neural network analysis. The authors also provide theoretical intuition by interpreting the critic update as a ridge-regression problem and arguing that the CKA regulariser effectively improves the conditioning of the optimisation problem by adding a positive semidefinite term to the normal matrix. While the analysis is simplified, it offers an interpretable perspective on why the regularisation might stabilise critic learning. The paper evaluates the proposed method across a range of commonly used reinforcement learning benchmarks and compares it with standard PPO as well as several stabilisation baselines. The experiments generally show improved stability and stronger final performance on several tasks, suggesting that the method may provide practical benefits in continuous-control settings. The paper is also generally well organised and structured, with the motivation, algorithmic modifications, and empirical results presented in a logical order that makes the overall narrative relatively easy to follow.

However, since the method combines multiple modifications, it is difficult to determine which component is primarily responsible for the observed improvements, and more comprehensive ablation studies would help clarify the individual effects of the critic regularisation, no-flip penalty, and exploration bonus. Additionally, the exploration component based on kernel density estimation introduces additional computational overhead and may not scale well to higher-dimensional state spaces.

---

> ### Author Rebuttal · Authors · 2026-03-29
>
> Thank you for the assessment and constructive comments. The questions and limitations are answered below.
>
> Q1
>
> Thank you for raising this point. Section 4.2 of the original manuscript already provides an ablation study over the three components, comparing full SPPO, variants with one component removed, and variants with only one component retained, with the corresponding RLiable metrics reported in Figure 7. The results show that each component alone improves PPO to some extent, while their combination usually performs better.
>
> The gain from CKA is more pronounced in the middle and later stages of training, where it mainly suppresses critic representation drift across updates and provides a more stable basis for subsequent improvement. The gain from No-Flip emerges earlier and remains beneficial throughout training, suggesting that it more directly constrains the directional consistency of actor updates. By contrast, KDE alone yields a milder gain, mainly in the early and middle stages, suggesting it is better viewed as an exploration-enhancing component rather than the main source of overall stabilization. More importantly, when KDE is combined with only one of the other two components, its gain remains limited. An explanation consistent with the observed results is that KDE changes exploration preferences through advantage shaping, but when only one side of the actor-critic pair is explicitly stabilized, this exploration benefit may not be effectively absorbed by the other side, and can therefore be partially offset. In contrast, when both CKA and No-Flip are present, the critic and actor are both explicitly stabilized, so the exploration signal introduced by KDE can be absorbed more effectively. This is why its benefit is more clearly expressed in full SPPO, especially through faster improvement in the early and middle stages.
>
> Functionally, CKA and No-Flip act more directly on critic-side and actor-side stability, respectively, while KDE mainly compensates for the potential loss of exploration induced by regularization. This is why the three are combined into a unified method. In the final version, the readability of Figure 7 will be improved, this conclusion will be stated more explicitly, and the above discussion will be revised accordingly.
>
> Q2
>
> Thank you for this question. The theoretical analysis in the main text and appendix is intended as a local mechanistic analysis under small updates, local linearization, and frozen normalization within the current update, rather than a strict global characterization of full deep nonlinear training dynamics. Under this approximation, the CKA term adds a positive semidefinite term to the normal equation, helping tighten the upper bounds on latent representation drift and one-step bootstrapping error. The empirical counterpart is reflected in Section 4.3, where SPPO shows more stable critic representation geometry and lower churn than PPO across multiple environments. Please also see the response to Reviewer J2H8, Q2, for a more detailed discussion of the assumptions, scope, and practical interpretation of this analysis. In the final version, this theory-empirics relationship will be further clarified.
>
> Q3
>
> Thank you for pointing out this issue. The KDE component does introduce additional computational cost. For Humanoid, the wall-clock time is about 2 hours for PPO and about 3 hours for SPPO; Appendix B.6 currently contains a typo on this point. The difference mainly comes from the KDE. In other words, we do not regard KDE as a zero-cost component.
>
> At the method level, KDE is applied in the critic latent space rather than the raw state space, so it is not directly tied to the raw state dimension. The motivation is to avoid the lack of semantic meaning in direct density estimation in the raw state space, and instead define novelty using critic latent representations that are more aligned with value-relevant semantics, which also alleviates dimensionality issues to some extent. Its efficiency remains affected by the latent dimension and the number of online samples, so it is primarily regarded as a practical mechanism to enhance exploration in the current continuous-control setting. In the final version, the additional runtime of KDE across environments and its dependence on latent dimensionality will be clarified more explicitly.
>
> Limitation 1
>
> This concern is essentially the same as Q3 above and will be made more explicit in the final version.
>
> Limitation 2
>
> Stronger CKA regularization is not always better, but reflects a trade-off between stability and adaptability, especially when the policy distribution changes substantially. This is also consistent with the discussion in Q2 above and the sensitivity results in Appendix C.6.
>
> Limitation 3
>
> No-Flip may introduce conservative bias when advantage estimates are noisy. Its effect should therefore be understood as a trade-off between stability and plasticity, rather than an unconditionally beneficial constraint.

---

> > ### Author Rebuttal · Reviewer_cyHV · 2026-04-02
> >
> > I thank authors for their comments. Their response is satisfactory and I am happy to keep my scores.

---

> > > ### Author Response · Authors · 2026-04-03
> > >
> > > Thank you for the positive feedback and for carefully considering the rebuttal. The maintained score is greatly appreciated.

---

### Decision · Program_Chairs · 2026-04-30

**Decision:**

Accept (regular)

**Comment:**

The paper introduces a stabilized variant of PPO for continuous control tasks. It stabilizes training and improves sample efficiency using three mechanisms: a constraint on critic representations, a "no-flip" penalty on actor updates, and KDE-based advantage shaping to encourage exploration.

Strengths:
- Extensive empirical evaluation across diverse benchmarks (MuJoCo, DMC, Atari) and multiple baselines.
- Theoretical grounding that mathematically bounds representation drift and bootstrapping error.
- Thorough ablation studies demonstrating the specific value of the proposed components.

Weaknesses:
- Large number of hyper-parameters to tune.
- Theoretical analysis relies on strong practical assumptions, such as local linearization and small update steps.
- KDE component introduces computational overhead and potential scaling concerns.

Given the empirical performance, theoretical motivation, and that all reviewers are satisfied with the paper, I recommend acceptance.